



# Cloud_cci AVHRR-PM dataset version 3: 35 year climatology of global cloud and radiation properties

Martin Stengel[1], Stefan Stapelberg[1,2], Oliver Sus[1,2], Stephan Finkensieper[1], Benjamin Würzler[1], Daniel Philipp[3,1], Rainer Hollmann[1], Caroline Poulsen[4], Matthew Christensen[4,5], and Gregory McGarragh[5]

[1]Deutscher Wetterdienst, Frankfurter Str. 135, 63067, Offenbach, Germany
[2]European Organisation for the Exploitation of Meteorological Satellites, Darmstadt, Germany
[3]Institute for Atmospheric and Environmental Sciences, Goethe University, Frankfurt am Main, Germany
[4]Science technology Facility Council, Rutherford Appleton Laboratory, Didcot, Oxfordshire, UK
[5]Department of Physics, University of Oxford, Clarendon Laboratory, Parks Road, Oxford OX1 3PU, UK

*Correspondence to:* Martin Stengel (martin.stengel@dwd.de)

**Abstract.** We present version 3 of the Cloud_cci AVHRR-PM dataset which contains a comprehensive set of cloud and radiative flux properties on a global scale covering the period of 1982 to 2016. The properties were retrieved from Advanced Very High Resolution Radiometer (AVHRR) measurements recorded by the afternoon (post meridiem, PM) satellites of the National Oceanic and Atmospheric Administration (NOAA) Polar Operational Environmental Satellites (POES) missions. The

5 cloud properties in version 3 are of improved quality compared with the precursor dataset version 2, providing better global quality scores for cloud detection, cloud phase and ice water path based on validation results against A-Train sensors. Furthermore, the parameter set was extended by a suite of broadband radiative flux properties. They were calculated by combining the retrieved cloud properties with thermodynamic profiles from reanalysis and surface properties. The flux properties comprise upwelling and downwelling, shortwave and longwave broadband fluxes at the surface (bottom-of-atmosphere - BOA) and

10 top-of-atmosphere (TOA). All fluxes were determined at AVHRR pixel level for all-sky and clear-sky conditions, which will particularly facilitate the assessment of the cloud radiative effect at BOA and TOA in future studies. Validation of the BOA downwelling fluxes against the Baseline Surface Radiation Network (BSRN) show a very good agreement. This is supported by comparisons of multi-annual mean maps with NASA's Clouds and the Earth's Radiant Energy System (CERES) products for all fluxes at BOA and TOA. The Cloud_cci AVHRR-PM version 3 dataset allows for a large variety of climate applications

15 that build on cloud properties, radiative flux properties and/or the link between them.

For the presented Cloud_cci AVHRR-PMv3 dataset a Digital Object Identifier has been issued:

https://doi.org/10.5676/DWD/ESA_Cloud_cci/AVHRR-PM/V003 (Stengel et al., 2019).



# 1 Introduction

Clouds play a critical role for the Earth's weather and climate through their contribution to the Earth's water cycle and their impact on the Earth's energy budget. Clouds impact the energy budget through their interaction with radiation, i.e. clouds usually reflect more solar radiation back to space than the underlying surface and absorb and re-emit infrared (IR) radiation,

leading to less IR radiation leaving the system than without clouds. Thus clouds significantly alter important components of the Earth's radiation budget: the shortware and longwave broadband fluxes at the top-of-atmosphere (TOA) and at the surface (bottom-of-atmosphere - BOA hereafter). Analysing cloud coverage and properties, and quantifying the impact they have on the radiation budget is of crucial importance for understanding the Earth's climate and the potential feedback mechanisms in a changing climate.

Since the beginning of the meteorological satellite era at the end of the 1970's attempts have been made to construct global cloud climatologies (e.g. Schiffer and Rossow, 1983) that are of sufficient quality to enable climate studies. Until recently the measurement records of metorological satellite sensors have grown now cover more than 40 years. Even though many difficulties exist when attempting to construct homogeneous and stable climate datasets, those multi-decadal satellite measurements provide the single most important source of measurements with global coverage. Some international efforts exist to

regularly improve and extend long-term satellite-based climatologies that contain a comprehensive suite of cloud properties: The Pathfinder Atmospheres - Extended (PATMOS-x, Heidinger et al., 2014), The International Satellite Cloud Climatology Project (ISCCP, Young et al., 2018), The EUMETSAT Climate Monitoring Satellite Application Facility (CM SAF) cloud and radiation data record (CLARA-A2, Karlsson et al., 2016) and the European Space Agency funded Climate Change Initiative ECV Cloud project (Cloud_cci, Stengel et al., 2017). All of these climatologies make use of measurements of the Advanced

Very High Resolution Radiometer (AVHRR), which is a passive imaging sensor with 5-6 spectral bands in the visible, near-infrared and thermal infrared. It is flown on the National Oceanic and Atmospheric Administration (NOAA) Polar Operational Environmental Satellites (POES) and on the EUMETSAT Meteorological operational satellite (Metop) series. There are newer passive sensors in space that also allow for constructing cloud datasets. These are part of research satellite missions by ESA (e.g. the (Advanced) Along-Track Scanning Radiometers on-board the European Remote Sensing Satellite (ERS-1/2) and the

Environmental Satellite (Envisat)) and by the National Aeronautics and Space Administration (NASA) (e.g. Moderate Resolution Imaging Spectroradiometer (MODIS) on board the Terra and Aqua satellites). However, mentioned research missions are often characterized by a significantly shorter data record and less spatial coverage due to smaller swath widths.

The MODIS record however has available the combination of cloud properties with high quality TOA radiation measurements made by the Clouds and the Earth's Radiant Energy System (CERES) sensors mounted on board the Terra and Aqua

satellites. In addition to the TOA radiation measurements, CERES BOA radiative fluxes are available based on simulations (Kato et al., 2013). Together with available clear-sky fluxes, this set-up provides an excellent basis for analysing the radiative effect of clouds on TOA and BOA energy balances, although the MODIS and CERES records exist only from the year 2000 onwards. Limitations might arise from the coarse spatial resolution of CERES (footprint size of approximately 30 km) and from the fact that the clear-sky fluxes are exclusively based on clear-sky pixels (and interpolation of clear-sky fluxes for gap





filling on monthly scales), for which the spatio-temporal sampling and the meteorological conditions are likely to be different. Consequently, small scale cloud processes and their radiative effects might not be resolved.

The World Climate Research Programme's (WCRP) Global Energy and Water Exchanges (GEWEX) Surface Radiation Budget (SRB) dataset is generated by application of a different approach. Here, retrieved cloud properties are used together

with reanalysis information and additional radiative transfer calculations in order to determine all-sky and clear-sky fluxes at the same time for each pixel. The latest release of the GEWEX SRB dataset (v3.0), however, only covers a period until 2007. It makes use of ISCCP DX data, which provides information on a temporal resolution of 3 hours, but includes some deficiencies such as utilizing less spectral information compared to AVHRR-based data, and a relatively coarse spatial resolution. The GEWEX SRB data have been used to revisit the cloud radiative effect on the global scale (e.g. Allan, 2011).

Based on the rationale above it seems logical to construct a record that includes both cloud and radiation properties based on AVHRR, covers a longer time period than alternative records, provides information at finer spatial scales (about 5 km for AVHRR global area coverage - GAC - data) and makes use of all 5 available spectral bands from the visible through the near-infrared to the thermal infrared. The usefulness of these data is further enhanced by the incorporation of the latest AVHRR intercalibration information and cloud retrieval developments.

This paper documents the approaches that have been followed to generate such an AVHRR-based data record with cloud and broadband radiative flux properties, and discusses derived results. The dataset is named Cloud_cci AVHRR-PMv3 (v3 hereafter) and is a successor of AVHRR-PMv2 (v2 hereafter), which contained cloud properties for the period 1982-2014 (see Stengel et al. (2017) for more details), and was already used in numerous studies, e.g. in model evaluation on the global scale (Lauer et al., 2017; Stengel et al., 2018; Eliasson et al., 2018) and on regional scales (Keller et al., 2018; Baró et al., 2018).

Superior to AVHRR-PMv2, AVHRR-PMv3 covers a longer time period (1982-2016), holds cloud properties of improved quality and includes broadband radiative flux properties at TOA and BOA. Appendx A lists additional information about the AVHRR measurement record used. To estimate the radiative fluxes additional radiative transfer calculations were conducted that included additional reanalysis information of tropospheric profiles of temperature and gaseous components as well as surface properties (all interpolated to AVHRR temporal and spatial resolution). This approach is similar to the GEWEX SRB

data, thus the retrieved cloud properties are ingested into the reanalysis profiles to represent real clouds with realistic properties at the correct time and place. This is considered a superior approach compared to using reanalysis (thus modelled) clouds directly. All of this information is then input to calculate the broadband fluxes. Although a considerable amount of reanalysis data is still required, this approach provides a means for quantifying the impact of true (retrieved) cloud properties on radiative fluxes at TOA and BOA in a realistic way. This also enables the collection of clear-sky fluxes at the same temporal frequency

as all-sky fluxes as opposed to collecting and interpolating the clear-sky fluxes into cloudy areas as is done for the CERES datasets.

In this paper the Cloud_cci AVHRR-PMv3 dataset is summarized. The following Section 2 reports recent cloud retrieval developments and updates, shows product examples and presents validation results all incorporating equivalent results from the precursor dataset version (v2). Section 3 introduces the radiative flux properties and the algorithms they are based on, and,

as for cloud properties, presents product examples and evaluation results. Section 4 gives a summary.





## 2 Cloud properties

The set of cloud properties included in v3 is identical to v2 and is outlined in the upper part of Table 1. All data are collected on two processing levels: (a) Level-3U which represents daily composites of non-averaged data collected on a global latitude-longitude grid with $0.05°$ resolution and (b) Level-3C which represents monthly averages and monthly histograms on a global latitude-longitude grid with $0.5°$ resolution. Input to Level-3U and Level-3C products are pixel-based retrievals using the algorithms described below. Further Level-3U and Level-3C specifications, i.e. the separation of data into liquid/ice sublayers as well as the histograms binning remain identical to v2 (see Tables 4 and 5 of Stengel et al. (2017)). The propagation of derived pixel-level uncertainties into the higher level products Level-3U and Level-3C remains identical to Stengel et al. (2017) as well.

### 2.1 Algorithms

The retrieval system employed for cloud properties is the Community Cloud retrieval for CLimate (CC4CL), which is summarized in Stengel et al. (2017) and described in detail in Sus et al. (2018) and McGarragh et al. (2018). However, further developments have taken place since v2, of which the key elements are listed in the following paragraphs. These improvements are grouped according to the CC4CL subcomponents: cloud masking and cloud phase determination, which now both employ Artificial Neural Network (ANN) schemes and require spectral band adjustments (SBAs), and a component for retrieving the remaining cloud properties using an optimal estimation technique (e.g. Rodgers, 2000).

- **Cloud mask**: The ANN for cloud detection ($ANN_{mask}$) has been retrained using a much larger set of training data, which is composed of collocations between AVHRR measurements and cloud optical depth observed by the Cloud-Aerosol Lidar with Orthogonal Polarization (CALIOP, Winker et al., 2009). In addition, a different set of channels for daytime conditions, the 3.7 $\mu m$ channel is now included in the ANN scheme (exception: 1.6 $\mu m$ is used for NOAA-16 for the period 04/2001 through 04/2003). Table B1 summarizes the $ANN_{mask}$ input data as a function of illumination conditions, while Table B2 reports the empirical thresholds applied posterior to convert the $ANN_{mask}$ output to a binary cloud mask. Downstream, cloud detection is complemented by an additional cirrus test based on 10.8 $\mu m$ and 12.0 $\mu m$ IR measurements as defined in Pavolonis et al. (2005). As the cloud detection was developed and fine-tuned for AVHRR on board the NOAA-19 satellites, SBAs are applied for other sensors, which is described in Appendix C. Cloud detection improvements compared to v2 are mainly found for daytime and twilight conditions in general, but in particular also for conditions with snow or ice covered surfaces and in cases of low-level liquid clouds over the sub-tropical and tropical oceans. Validation scores are presented in Section 2.3 reflecting the improvements on the global scale.

- **Cloud top phase** determination, which in v2 was inferred from the cloud typing procedure of Pavolonis and Heidinger (2004) and Pavolonis et al. (2005), was replaced by a ANN approach for v3 ($ANN_{phase}$). The strategy for training the $ANN_{phase}$ was very similar compared to the cloud detection approach, training the $ANN_{phase}$ to emulate CALIOP cloud top phase using AVHRR measurements as primary input data. The exact list of input data for the $ANN_{phase}$ is given in Table B3. Table B4 lists the thresholds applied to convert the $ANN_{phase}$ output to a binary cloud phase. As for cloud





**Table 1.** Cloud_cci AVHRR-PMv3 cloud and radiation properties. $ANN_{mask}$ = artificial neural network for cloud detection, $ANN_{phase}$ = artificial neural network for cloud phase, SV = state vector, PP = postprocessed, PV = Pavolonis algorithm (Pavolonis and Heidinger, 2004; Pavolonis et al., 2005), OE = optimal estimation, BR = BUGSrad (radiative flux algorithm), TOA = top-of-atmosphere, BOA = bottom-of-atmosphere (surface), LW = longwave, SW = shortwave. Upper part of the table (cloud properties) has been adopted from Sus et al. (2018).

| Variable name | Abbreviation | Unit | Origin | Comment |
|---|---|---|---|---|
| Cloud properties | | | | |
| cloud mask / | CMA / | 1 | $ANN_{mask}$ | binary cloud occurrence classification / |
| cloud fraction | CFC | % | | Fraction of cloudy pixels |
| cloud phase / | CPH / | 1 | $ANN_{phase}$ | binary cloud phase classification / |
| liquid cloud fraction | LCF | % | | Fraction of liquid clouds |
| cloud top pressure | CTP | hPa | SV | OE retrieval result of cloud top pressure |
| cloud top height | CTH | km | PP | derived from CTP and atmospheric profile |
| cloud top temperature | CTT | Kelvin | PP | derived from CTP and atmospheric profile |
| cloud effective radius | CER | $\mu$m | SV | OE retrieval result of cloud effective radius |
| cloud optical thickness | COT | 1 | SV | OE retrieval result of cloud optical thickness |
| surface temperature | STEMP | Kelvin | SV | OE retrieval result of surface temperature |
| cloud water path | CWP | g m$^{-2}$ | PP | derived from CER and COT (Stephens, 1978) |
| cloud albedo at 0.6 $\mu$m | $CLA_{0.6}$ | 1 | PP | derived from CER and COT |
| cloud albedo at 0.8 $\mu$m | $CLA_{0.8}$ | 1 | PP | derived from CER and COT |
| cloud effective emissivity | CEE | 1 | PP | derived from 10.8 and 12.0 $\mu$m data |
| Broadband flux properties | | | | |
| TOA upwelling SW flux | $SWF_{TOA}^{up}$, clear$SWF_{TOA}^{up}$ | W m$^{-2}$ | BR | all-sky and clear-sky |
| TOA upwelling LW flux | $LWF_{TOA}^{up}$, clear$LWF_{TOA}^{up}$ | W m$^{-2}$ | BR | all-sky and clear-sky |
| BOA upwelling SW flux | $SWF_{BOA}^{up}$, clear$SWF_{BOA}^{up}$ | W m$^{-2}$ | BR | all-sky and clear-sky |
| BOA upwelling LW flux | $LWF_{BOA}^{up}$, clear$LWF_{BOA}^{up}$ | W m$^{-2}$ | BR | all-sky and clear-sky |
| BOA downwelling SW flux | $SWF_{BOA}^{down}$, clear$SWF_{BOA}^{down}$ | W m$^{-2}$ | BR | all-sky and clear-sky |
| BOA downwelling LW flux | $LWF_{BOA}^{down}$, clear$LWF_{BOA}^{down}$ | W m$^{-2}$ | BR | all-sky and clear-sky |
| Photosynthetic Active Radiation | PAR | W m$^{-2}$ | BR | total and diff |

detection, SBAs are applied prior to the cloud phase determination (see Appendix C). Significant improvements are found for the cloud phase in v3 compared to v2 when analysing validation results against CALIOP as reported in Section 2.3.

- **OE retrieval of cloud properties**: The surface reflectance model was revised leading to a corrected handling of the solar zenith angle with most pronounced changes at large angles. Furthermore, bugs were fixed in the code that composes the





look-up tables (LUTs) based on pre-calculated radiative transfer simulations. In particular the LUTs for channels with solar reflectance contribution changed considerably. This led to smaller CER retrievals for $3.7\mu m$ measurements, in particular for $CER_{ice}$. Introducing the utilization of the ice cloud single-scattering properties of Baum et al. (2014) (Baran et al. (2005) used before) further reduced the $CER_{ice}$. For AVHRR-PMv3 cloud optical properties are also retrieved dur-

ing night-time, facilitated by a differential sensitivity of the radiation in the spectral bands $3.7\mu m$ and $10.8\mu m/12.0\mu m$ to COT and CER. Night-time COT and CER retrievals are considered as experimental products and only included in Level-3U products. All retrieved cloud properties are input to the calculation of the radiative fluxes as described in Section 3. As for v2, retrievals of COT and CER are used in v3 to determine LWP and IWP following Stephens (1978).

## 2.2    Cloud property examples

Figure 1 shows global maps of monthly mean CFC, LCF, COT and CER for June 2014 for v3 Level-3C data - along with the same data from v2. In general, global patterns look very similar with only minor differences between v3 and v2 for CFC and COT. LCF increased (more liquid clouds) from v3 to v2 after a fundamental change of the phase detection approach (see above). CER of v3 is significantly lower than in v2, which is mainly due to fixing a bug in some CC4CL LUTs and introducing alternative single scattering properties, as mentioned in Section 2.1, which only affected retrieved ice cloud properties.

Figure 2 presents the same comparison for CTP, LWP, IWP and $CLA_{0.6}$. Global patterns remain very similar again. Mean CTP is lower in v3 than in v2 in the Tropics, which is predominantly due to detecting more very-low level clouds above tropical oceans. While LWP remains similar in v3 compared to v2, IWP is significantly lower in v3 due to lower $CER_{ice}$ (input to the IWP calculation). Unrealistically high LWP and IWP values in polar regions are reduced in v3 due to reduced CER. $CLA_{0.6}$ is slightly higher in v3 compared to v2 although the changes are relatively small.

Detailed validation was carried out for all cloud properties for which accurate reference data exist. The results of those efforts are presented in the next section, highlighting the quality of the v3 data.

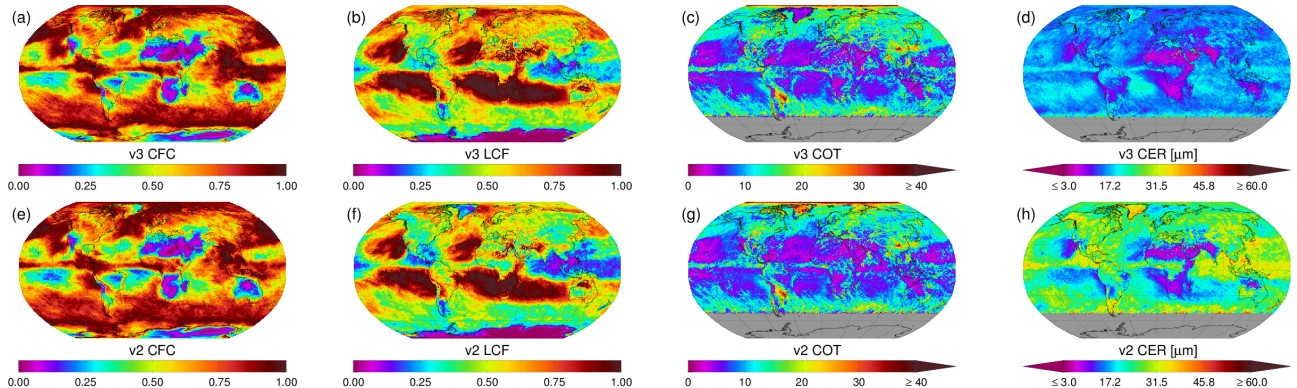

**Figure 1.** Examples of Level-3C (monthly means) Cloud_cci AVHRR-PMv3 data for cloud fraction CFC (a), liquid cloud fraction LCF (b), cloud optical thickness COT (c) and cloud effective radius CER (d). Same data is shown for v2 (e-h). All data is from June 2014.

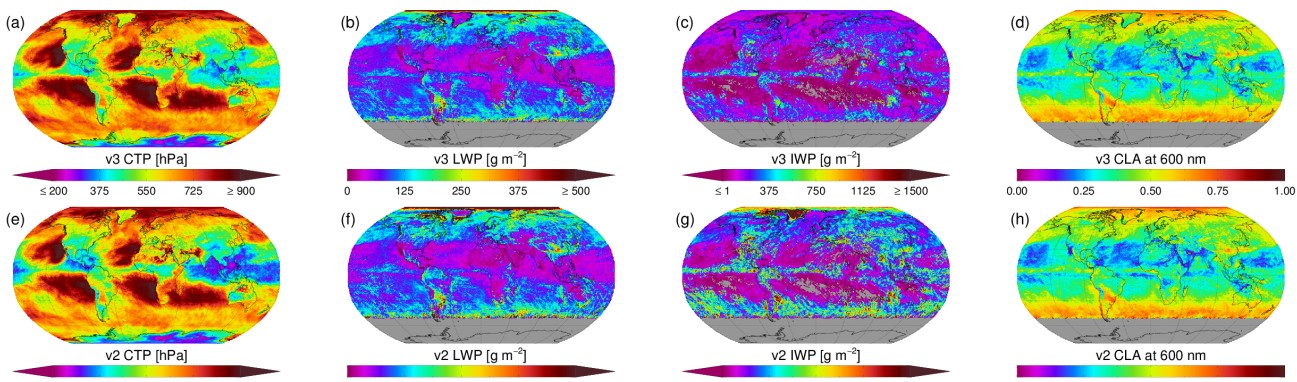

**Figure 2.** As Figure 1 but for CTP, LWP, IWP and CLA

## 2.3 Validation

Cloud_cci AVHRR-PMv3 CMA, CPH and CTH Level-3U products were collocated with equivalent CALIOP products which are assumed to be of superior quality. More specifically, the CAL_LID_L2_05kmCLay-Prov product was downloaded from the ICARE Data and Service Center (http://www.icare.univ-lille1.fr). To investigate the sensitivity of passive imager retrievals to the thinnest cloud layers, the cloud optical depth profiles included in the CALIOP profiles were employed as in Karlsson and Johansson (2013); Stengel et al. (2013); Sus et al. (2018). Following this approach different scenarios for excluding optically thin cloud layers are investigated when discussing validation of CMA, CPH and CTH below.

In addition to the validation against CALIOP, Cloud_cci AVHRR-PMv3 LWP was collocated with AMSR-E observations of LWP (Wentz and Meissner, 2004), and IWP was collocated to DARDAR (raDAR/liDAR Delanoë and Hogan, 2008, 2010) observations of IWP. Passive microwave observations of AMSR-E over ocean and active observations of CALIOP and Cloud-sat in DARDAR are assumed to provide best reference data for LWP and IWP on global scales. All validation results are accompanied by the equivalent results for v2.

Table 2 reports the validation results for CMA for two scenarios: (1) considering all CALIOP reference pixels as cloudy for which the CALIOP COT is above 0.0 ($COT_{thres}$=0.0), and (2) considering only those CALIOP reference pixels as cloudy for which the CALIOP COT is above 0.15 ($COT_{thres}$=0.15). The latter scenario is added to account for the lack of sensitivity of AVHRR measurements to very optically thin clouds. For both scenarios, the scores are generally better for v3 than for v2. Heidke Skill Scores (Heidke, 1926), hitrates and probabilities of detections (PODs) are higher (thus better). The only degradation in the scores is found for the bias, which is slightly more negative in v3 compared to v2.

Table 3 reports the validation results for CPH, also for two scenarios: (1) using the cloud phase at the top of the uppermost cloud layer detected by CALIOP as reference ($COT_{lev}$=0.0), and (2) using the cloud phase at an optical depth of 0.15 into the cloud (top-down) as reference ($COT_{lev}$=0.15). Comparing the HSS score as overall measure for the correct cloud phase detection, v3 performs better than v2. This is mainly caused by an improved identifications of liquid clouds which comes at the cost of a slightly reduced POD for ice clouds. The liquid bias has increased for v3. Removing the thinnest cloud layers,




thus accounting for the AVHRR sensor limitation, the improvement of v3 over v2 becomes even clearer. In this scenario, the cloud phase of 84.7 % of all clouds is correctly identified in v3 (according to hitrate scores). It is important to note that the CALIOP data used for validation of cloud detection and cloud phase determination excluded the data that was used for training the ANNs.

Table 4 reports the validation results for CTH. The validation is stratified by the phase of the cloud and by optical depth into the cloud (top-down) at which the reference CTH is taken from the CALIOP profile. In addition to $COT_{lev}$ of 0.0 and 0.15, $COT_{lev}$ of 1.0 is also included. Generally only few changes in validations scores are found between v3 and v2. While for liquid clouds the scores remain nearly the same, a small degradation in the CTH bias for ice clouds is found. The underestimation of CTH is stronger in v3 compared to v2. For example for the geometrical CTH from CALIOP ($COT_{lev}$=0.0) the bias degrades

from -2.59 km to -3.54 km. One reason for this can be that the LUT-related bug fixes (see Section 2.1) led to smaller $CER_{ice}$. Smaller ice particles absorb less radiation coming from below the cloud putting the cloud lower in the atmosphere in the retrieval. In contrast to the bias, standard deviations are reduced for v3 amounting to 2.36 km compared to 2.51 km in v2. For $COT_{lev}$=0.15 and $COT_{lev}$=1.0, very similar findings are made with both of these scenarios showing the reduction in bias and standard deviation with increasing $COT_{lev}$ for ice clouds. This highlights the difficulties in correctly placing (vertically)

optically thin clouds and cloud layers when using AVHRR measurements.

    Table 5 reports the validation results for LWP. Although the bias for v3 remains small when compared against AMSR-E, it is slightly increased compare to v2 from -1.9 g m$^{-2}$ to -3.2 g m$^{-2}$. Standard deviations are slightly decreased for v3 (26.4 g m$^{-2}$) compared to v2 (27.1 g m$^{-2}$) and the correlation remains unchanged at 0.65.

    Table 6 reports the validation results for IWP. The AVHRR-PM IWP generally shows an underestimation of IWP when

DARDAR is considered as reference. This underestimation has increased for v3 as the bias has become larger and negative (-307.1 g m$^{-2}$ for v3 compared to 33.3 g m$^{-2}$ for v2). However, the standard deviation has decreased significantly from 1299.8 g m$^{-2}$ to 900.9 g m$^{-2}$ along with a clear increase in correlation from 0.42 to 0.63.

    Despite the assumption that the reference data used is of higher quality than the Cloud_cci data, uncertainties and inaccuracies remain in the reference data as well, which should be kept in mind when interpreting the presented validation scores.

However, summarizing the discussion above, the cloud properties included in AVHRR-PMv3 are considered to be of superior quality than the precursor version.



**Table 2.** Cloud mask (CMA) validation results for Cloud_cci AVHRR-PMv3 when compared against CALIOP. Validation results for AVHRR-PMv2 are also reported. Validation measures are the probabilities of detecting cloudy and clear scenes (Hitrate, $POD_{cloudy}$, $POD_{clear}$) and bias. In addition, the number of collocated pixels is given. The scores are separated into two cloud optical thickness thresholds ($COT_{thres}$) reflecting above which CALIOP COT the CALIOP pixel was classified cloudy.

| | Score | AVHRR-PMv3 | AVHRR-PMv2 |
|---|---|---|---|
| $COT_{thres} = 0.0$ | HSS | 0.68 | 0.64 |
| | Hitrate [%] | 79.23 | 78.17 |
| | $POD_{cloudy}$ [%] | 75.82 | 75.46 |
| | $POD_{clear}$ [%] | 91.98 | 88.31 |
| | Bias [%] | -17.38 | -16.89 |
| | Number | 16 139 764 | 16 139 764 |
| $COT_{thres} = 0.15$ | HSS | 0.66 | 0.63 |
| | Hitrate [%] | 83.01 | 81.86 |
| | $POD_{cloudy}$ [%] | 83.29 | 82.79 |
| | $POD_{clear}$ [%] | 82.45 | 79.97 |
| | Bias [%] | -5.35 | -4.86 |
| | Number | 16 139 764 | 16 139 764 |



**Table 3.** Cloud phase (CPH) validation results for Cloud_cci AVHRR-PMv3 when compared against CALIOP. Validation results for AVHRR-PMv2 are also reported. Validation measures are the probabilities of detecting liquid and ice phase (Hitrate, $POD_{liq}$, $POD_{ice}$ and bias of liquid cloud occurrence). In addition the number of collocated pixels is given. The scores are separated into two cloud optical depth levels ($COT_{lev}$) representing at which top-down COT into the cloud the reference CALIOP CPH was taken.

| | Score | AVHRR-PMv3 | AVHRR-PMv2 |
|---|---|---|---|
| $COT_{lev} = 0.0$ | HSS | 0.62 | 0.56 |
| | Hitrate [%] | 79.74 | 77.87 |
| | $POD_{liq}$ [%] | 86.25 | 78.02 |
| | $POD_{ice}$ [%] | 75.44 | 77.77 |
| | Bias [%] | 9.35 | 4.67 |
| | Number | 8 788 655 | 8 788 655 |
| $COT_{lev} = 0.15$ | HSS | 0.69 | 0.62 |
| | Hitrate [%] | 84.70 | 80.99 |
| | $POD_{liq}$ [%] | 82.33 | 74.06 |
| | $POD_{ice}$ [%] | 87.16 | 88.20 |
| | Bias [%] | -2.72 | -7.44 |
| | Number | 8 435 631 | 8 435 631 |





**Table 4.** Cloud-top height (CTH) validation results for Cloud_cci AVHRR-PMv3 when compared against CALIOP. Validation results for AVHRR-PMv2 are also reported. Validation measures are standard deviation (Std) and bias. In addition the number of collocated pixels is given. All scores are separated into liquid and ice clouds (both Cloud_cci dataset and CALIOP had to agree on phase) and into three cloud optical depth levels ($COT_{lev}$) representing at which top-down COT into the cloud the reference CALIOP CTH was taken.

|  | Score | **AVHRR-PMv3** | **AVHRR-PMv2** |
|---|---|---|---|
| $COT_{lev} = 0.0$ | $Std_{liq}$ [km] | 0.86 | 0.86 |
|  | $Bias_{liq}$ [km] | -0.10 | -0.11 |
|  | $Number_{liq}$ | 2 603 163 | 2 603 163 |
|  | $Std_{ice}$ [km] | 2.36 | 2.51 |
|  | $Bias_{ice}$ [km] | -3.54 | -2.59 |
|  | $Number_{ice}$ | 3 691 179 | 3 691 179 |
| $COT_{lev} = 0.15$ | $Std_{liq}$ [km] | 0.91 | 0.91 |
|  | $Bias_{liq}$ [km] | -0.06 | -0.08 |
|  | $Number_{liq}$ | 3 016 985 | 3 016 985 |
|  | $Std_{ice}$ [km] | 2.14 | 2.30 |
|  | $Bias_{ice}$ [km] | -2.95 | -2.00 |
|  | $Number_{ice}$ | 3 376 337 | 3 376 337 |
| $COT_{lev} = 1.0$ | $Std_{liq}$ [km] | 0.80 | 0.80 |
|  | $Bias_{liq}$ [km] | 0.05 | 0.04 |
|  | $Number_{liq}$ | 2 982 690 | 2 982 690 |
|  | $Std_{ice}$ [km] | 1.95 | 2.09 |
|  | $Bias_{ice}$ [km] | -1.62 | -0.84 |
|  | $Number_{ice}$ | 2 077 074 | 2 077 074 |

**Table 5.** Liquid water path (LWP) validation results for Cloud_cci AVHRR-PMv3 when compared against AMSR-E for the year 2008. Validation results for AVHRR-PMv2 for the same time period are also reported. Validation measures are standard deviation (Std), bias and correlation. In addition the number of collocated pixels is given.

| Score | **AVHRR-PMv3** | **AVHRR-PMv2** |
|---|---|---|
| Std [g m$^{-2}$] | 26.4 | 27.1 |
| Bias [g m$^{-2}$] | -3.2 | -1.9 |
| Correlation | 0.64 | 0.64 |
| Number | 183 022 | 183 022 |





**Table 6.** Ice water path (IWP) validation results for Cloud_cci AVHRR-PMv3 when compared against DARDAR for January to July 2008. Validation results for AVHRR-PMv2 for the same time period are also reported. Validation measures are standard deviation (Std), bias and correlation. In addition the number of collocated pixels is given.

| Score | AVHRR-PMv3 | AVHRR-PMv2 |
|---|---|---|
| Std [g m$^{-2}$] | 900.9 | 1299.8 |
| Bias [g m$^{-2}$] | -307.1 | 33.3 |
| Correlation | 0.63 | 0.42 |
| Number | 92 293 | 92 293 |

## 3   Radiation properties

In addition to the cloud properties described in the previous section, radiative broadband flux properties (shortwave and longwave) at TOA and BOA, and for all-sky and clear-sky conditions, were calculated employing the BUGSrad scheme (Stephens et al., 2001, more details below). Furthermore, the photosynthetically active radiation was determined. A full list of radiation

properties is given in the bottom part of Table 1. As for the cloud properties, all radiation properties are derived at pixel level, sub sampled to daily, global composites (Level-3U products) and aggregated to monthly Level-3C products.

### 3.1   Algorithm

BUGSrad uses a two-stream approximation along with correlated-k distribution methods for atmospheric radiative transfer (Fu and Liou, 1992). It has been used to investigate aerosol-cloud interactions (Christensen et al., 2017) and to assess the

Earth's energy budget using CloudSat observations (Stephens et al., 2012). BUGSrad is applied to a single column, plane-parallel atmosphere with ingested cloud properties (i.e. CER, COT, CTP) previously retrieved with CC4CL (see Section 2.1). BUGSrad uses 18 spectral bands in the electromagnetic spectrum (6 in the shortwave and 12 in the longwave) to compute the broadband fluxes. Atmospheric profiles for temperature and water vapour are taken from ERA-Interim. Visible and near-infrared surface albedo are based on spatiotemporally resolved MODIS climatologies - with all data being identical to the usage

in CC4CL. Total solar irradiance is based on SOHO (Solar and Heliospheric Observatory) and SORCE (SOlar Radiation and Climate Experiment) measurements acquired from http://disc.sci.gsfc.nasa.gov/SORCE/data-holdingsusingSOR3TSID_v017 and further processed by applying a bi-linear interpolation followed by a bias correction to SOHO measurements to match SORCE. For well-mixed radiatively important trace gases constant values are used ($CH_4$ = 1.8 ppm, $N_2O$ = 0.26 ppm). For $CO_2$ a linearly time dependent concentration is used anchored at 380 ppm for the year 2006. To account for the effect of aerosols

on the radiation, an aerosol optical depth of 0.05 was added to the extinction throughout the atmosphere. It is acknowledged that this value is under-representing heavy aerosol loadings which motivates the utilization of spatiotemporally resolved aerosol information for future dataset versions.

Due to the angular dependence of the solar illumination together with the low sampling frequency of a single polar-orbiting AVHRR sensor, an angular dependence correction is applied to the shortwave radiation properties to make the data represent





24 hour averages. This is done by calculating the diurnal cycle of the solar zenith angle (SZA) for a given pixel on the day of observation. The diurnal cycle of SZA is then used to rescale the incoming and reflected solar radiation and the atmospheric path length for a given set of time stamps throughout the local day. Averaging these samples gives a suitable approximation for a true 24 hour mean, which is needed to determine true climatological means. This procedure is however only applied for

Level-3C products, while Level-3U products hold the instantaneous, uncorrected fluxes representing the solar illumination at the pixel location and at the time of observation.

For longwave radiation, a diurnal cycle correction is applied based on a cosine fit to an observed mean diurnal cycle by applying CC4CL to geostationary Spinning Enhanced Visible and Infrared Imager (SEVIRI). The observed diurnal cycle is converted to a correction factor, which itself is a function of local observation time, to mimic a 24 hour mean.

## 3.2 Radiation property examples

Figure 3 shows examples of Cloud_cci AVHRR-PMv3 Level-3C data of $SWF_{TOA}^{up}$, $LWF_{TOA}^{up}$, $SWF_{BOA}^{down}$, $LWF_{BOA}^{down}$ for all-sky and clear-sky conditions for June 2014. As a general description of these properties, high clear$SWF_{TOA}^{up}$ is found in regions with high surface albedo while high values in $SWF_{TOA}^{up}$ are additional visible in regions with high cloud fraction, and vice versa. Clear$SWF_{TOA}^{up}$ and $SWF_{TOA}^{up}$ depend on incoming solar flux, which, in the month of June, is highest in the tropics and Northern

Hemisphere. Clear$LWF_{TOA}^{up}$ is highest in regions with high surface temperatures and low water vapour amounts in the atmospheric column above. Higher water vapour loadings and in particular frequent occurrence of cold clouds significantly reduce the $LWF_{TOA}^{up}$, for example visible in the tropics and the mid-latitudes. $SWF_{BOA}^{down}$ represents the downwelling solar radiation that is neither reflected nor absorbed by clouds or the atmosphere, thus is, roughly speaking, high where $SWF_{TOA}^{up}$ is low and vice versa. $SWF_{BOA}^{down}$ and clear$SWF_{BOA}^{down}$ strongly depend on illumination conditions. $LWF_{BOA}^{down}$ represents the downwelling radiation

emitted by the atmosphere and clouds and is high in regions with high water vapour amounts and further increased when clouds are frequently present.

The product portfolio for radiative fluxes is complemented by $SWF_{TOA}^{down}$: the incoming solar radiation at the top of the atmosphere and $SWF_{BOA}^{up}$ ($LWF_{BOA}^{up}$): the reflected solar (emitted terrestrial) radiation at the Earth's surface (not shown).

### 3.3 Validation

### 3.3.1 BOA radiative fluxes

The Cloud_cci AVHRR-PMv3 BOA radiative fluxes $SWF_{BOA}^{down}$ and $LWF_{BOA}^{down}$ were compared against ground-based reference stations of the World Radiation Monitoring Center (WRMC) Baseline Surface Radiation Network (BSRN, Driemel et al., 2018). For this, monthly mean BSRN $SWF_{BOA}^{down}$ and $LWF_{BOA}^{down}$ values were calculated per station from all available observations and then compared to the nearest neighbouring Cloud_cci grid box. Figure 4 shows scatter plots for all monthly pairs found

within the period of 2003 to 2016. The validation scores are reported in Table 7. An excellent agreement of the Cloud_cci with the reference BSRN measurements is found for both $SWF_{BOA}^{down}$ and $LWF_{BOA}^{down}$ products with correlations above 0.98. Standard deviations are 13.8 W m$^{-2}$ for $SWF_{BOA}^{down}$ and 11.5 W m$^{-2}$ for $LWF_{BOA}^{down}$. The comparisons further reveal positive biases



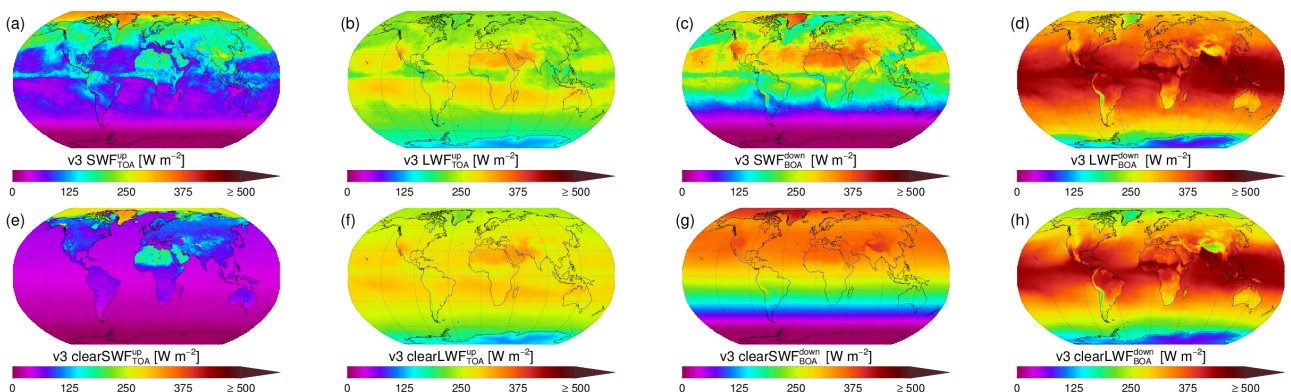

**Figure 3.** Examples of Level-3C (monthly means) Cloud_cci AVHRR-PMv3 data of $SWF_{TOA}^{up}$, $LWF_{TOA}^{up}$, $SWF_{BOA}^{down}$, $LWF_{BOA}^{down}$ (a-d). clearSWF$_{TOA}^{up}$, clearLWF$_{TOA}^{up}$, clearSWF$_{BOA}^{down}$, clearLWF$_{BOA}^{down}$ are shown in (e-h) representing the same fluxes as in (a-d) but for clear-sky conditions. All data is from June 2014.

in Cloud_cci: 1.9 W m$^{-2}$ for $SWF_{BOA}^{down}$ and 7.6 W m$^{-2}$ for $LWF_{BOA}^{down}$. Considering the $LWF_{BOA}^{down}$ bias, Nyeki et al. (2017) recently found indications that the measured fluxes at BSRN stations are biased low. They quantified this with 3.5 to 5.4 W m$^{-2}$ which has the potential to explain more than 50 % of the bias found between Cloud_cci and BSRN for $LWF_{BOA}^{down}$. Figure 4 also shows equivalent validation for upwelling fluxes at those BSRN sites which provide upwelling measurements (much fewer stations than for downwelling fluxes). For $LWF_{BOA}^{up}$ the agreement of Cloud_cci to BSRN is again very good with standard deviation of 14.1 W m$^{-2}$, a bias of -3.0 W m$^{-2}$ and a correlation of 0.99.

In general, the agreement of the Cloud_cci $SWF_{BOA}^{down}$, $LWF_{BOA}^{down}$ and $LWF_{BOA}^{up}$ with the BSRN stations is remarkable when considering that only one satellite sensor is used at a time, thus for many locations on Earth only two satellite overpasses (one daytime, one night-time) within 24 hours provide observations. The results are a confirmation that the developed and applied diurnal cycle correction works well, which is more important for the shortwave than for the longwave fluxes.

In contrast, for $SWF_{BOA}^{up}$ more scatter is found in the comparisons to BSRN. Considering $SWF_{BOA}^{up}$ is simply the $SWF_{BOA}^{down}$ multiplied with the surface albedo, and the good validation results for $SWF_{BOA}^{down}$, this leads to the conclusions that either imperfect surface albedo was used in Cloud_cci, or, more likely, the difference in spatial scales might be the dominating source of the discrepancy found. Fine scale inhomogeneities in surface albedo in the vicinity of the BSRN stations will propagate into the results.

In addition to the BSRN stations, Cloud_cci BOA downwelling and upwelling fluxes were compared to the Clouds and the Earth's Radiant Energy System (CERES) Energy Balanced and Filled (EBAF) surface flux product (Kato et al., 2013), i.e. by means of comparing multi-annual mean maps for the period 2003-2016 (Figures 5 and 6), with corresponding latitude-weighted global mean values given in Table 8.

The Cloud_cci multi-annual mean maps of $SWF_{BOA}^{down}$ for the chosen period agree very well with the CERES products (panels a and b of Figure 5), also for the clear-sky fluxes (panels c and d of Figure 5). This is also supported by global mean values





reported in Table 8 in which Cloud_cci is slightly biased high (+0.9 W m$^{-2}$ for SWF$_{BOA}^{down}$ and +2.2 W m$^{-2}$ for clearSWF$_{BOA}^{down}$). Clear-sky fluxes in both products are mainly characterized by larger incoming solar radiation at the equator, scattering and absorption by atmospheric gases and aerosols, and the surface reflectivity and emissivity. The presence of clouds usually leads to a significant reduction of SWF$_{BOA}^{down}$ locally being a function of optical thickness and cloud fraction over larger domains. The

fact that the all-sky fluxes SWF$_{BOA}^{down}$ agree very well with CERES validates the Cloud_cci cloud detection and corresponding cloud property retrievals, which thus can be assumed to be of high quality.

The Cloud_cci multi-annual mean maps of LWF$_{BOA}^{down}$ (Figure 6) also agree well with CERES in terms of global patterns. The absolute values however show systematically higher values for Cloud_cci of about 8 to 9 W m$^{-2}$ for both all-sky and clear-sky values. In relative terms the systematic differences amount to approximately 2 to 3 %. However, these differences lie within

the expected range of the CERES accuracy (Rutan et al., 2015).

The Cloud_cci multi-annual mean maps of SWF$_{BOA}^{up}$ exhibit larger systematic deviations (not shown) than for SWF$_{BOA}^{down}$. The larger standard deviations retrieved form the solar reflected radiation is primarily related to variances in surface albedo and cloud cover which tend to have significant annual cycles. Global mean values reported in Table 9 give negative biases of -2.7 and -4.6 W m$^{-2}$ for Cloud_cci which in relative terms correspond to negative deviations of more than 10 %. It remains uncertain

which of the two products are more realistic as no real ground truth is available for SWF$_{BOA}^{up}$ that represents spatial scales of satellite pixels (several kilometres). Repeating the validation of SWF$_{BOA}^{up}$ against BSRN but using CERES gives comparable, large deviations (not shown) as found for Cloud_cci (see above). This is in agreement with findings of (Kratz et al., 2010) who reported systematic deviations between CERES and surface observations of SWF$_{BOA}^{up}$ depending on time of day, meteorological condition and location.

Cloud_cci multi-annual maps of LWF$_{BOA}^{up}$ are again closer to CERES (not shown). Global mean values (Table 9) deviate by approximately 2 W m$^{-2}$ only with larger values for Cloud_cci. In relative terms the differences are about 0.5 %.

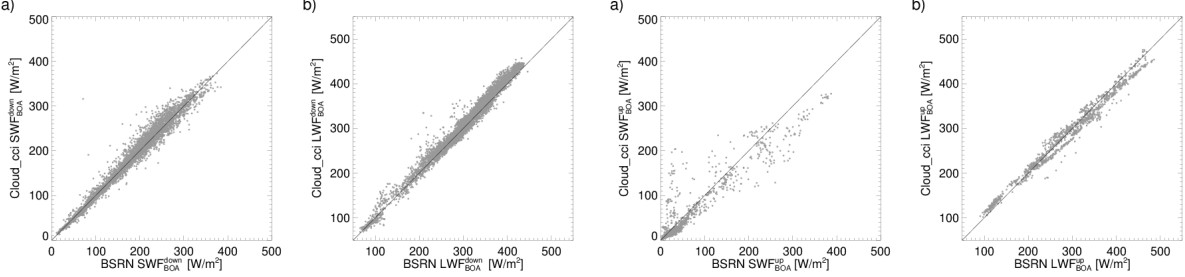

**Figure 4.** Comparison of Cloud_cci bottom-of-atmosphere (BOA) shortwave (SW, panel a) and longwave (LW, panel b) downwelling fluxes with ground-based reference measurements taken at globally distributed Baseline Surface Radiation Network (BSRN) sites for which equivalent reference data was available. Panels (c) and (d) are as (a) and (b) but for upwelling fluxes. Shown are all monthly data pairs within the period 2003-2016.




**Table 7.** Validation results for monthly Cloud_cci AVHRR-PMv3 shortwave and longwave, downwelling and upwelling radiative fluxes at bottom-of-atmosphere (BOA) when compared against Baseline Surface Radiation Network (BSRN) sites within the period 2003-2016. Validation measures are standard deviation (Std), bias and correlation. In addition the number of data pairs is given.

| Score | $SWF_{BOA}^{down}$ | $LWF_{BOA}^{down}$ | $SWF_{BOA}^{up}$ | $LWF_{BOA}^{up}$ |
|---|---|---|---|---|
| Std [W m$^{-2}$] | 13.83 | 11.52 | 31.18 | 14.11 |
| Bias [W m$^{-2}$] | 1.99 | 7.60 | -6.16 | -3.02 |
| Correlation | 0.98 | 0.99 | 0.93 | 0.99 |
| Number | 4487 | 5627 | 1022 | 1182 |

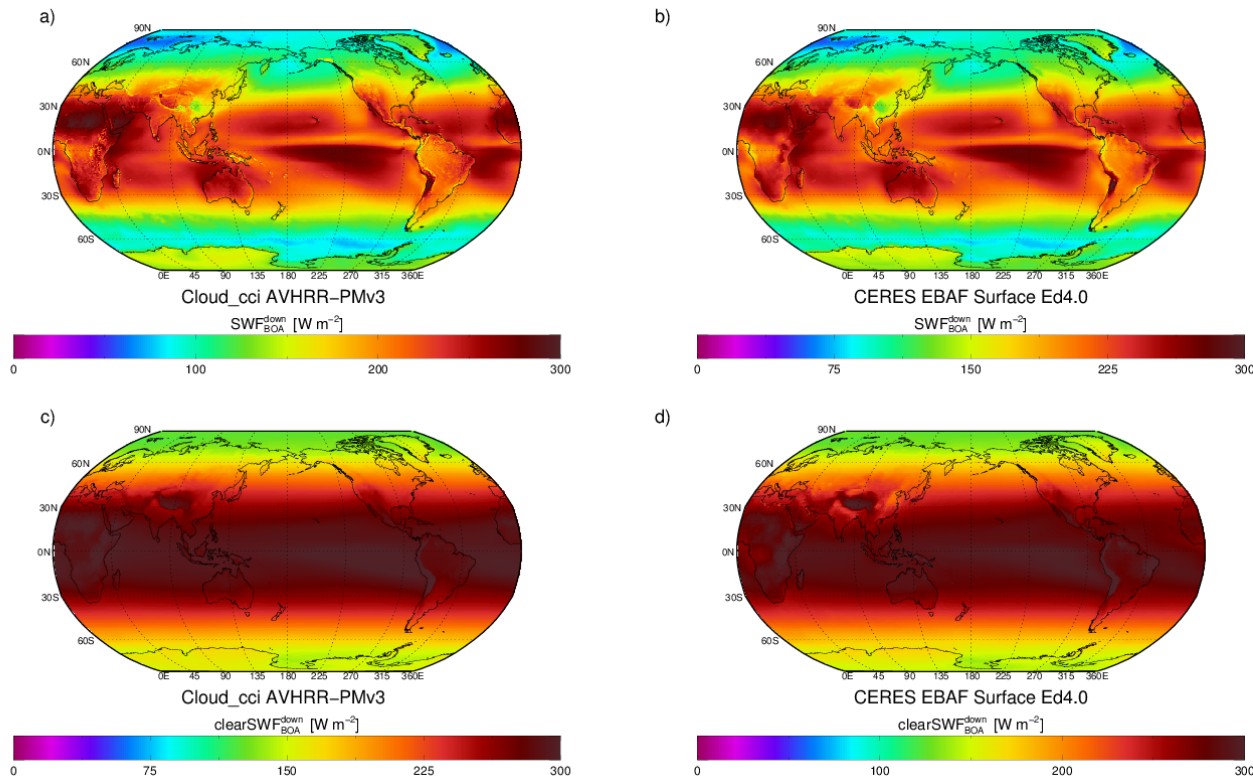

**Figure 5.** Multi-annual (2003-2016) mean downwelling shortwave (SW) radiative fluxes at bottom-of-atmosphere (BOA) for all-sky conditions for Cloud_cci AVHRR-PMv3 (a) and CERES EBAF surface fluxes (b). Panels (c) and (d) show the same data but for clear-sky conditions.



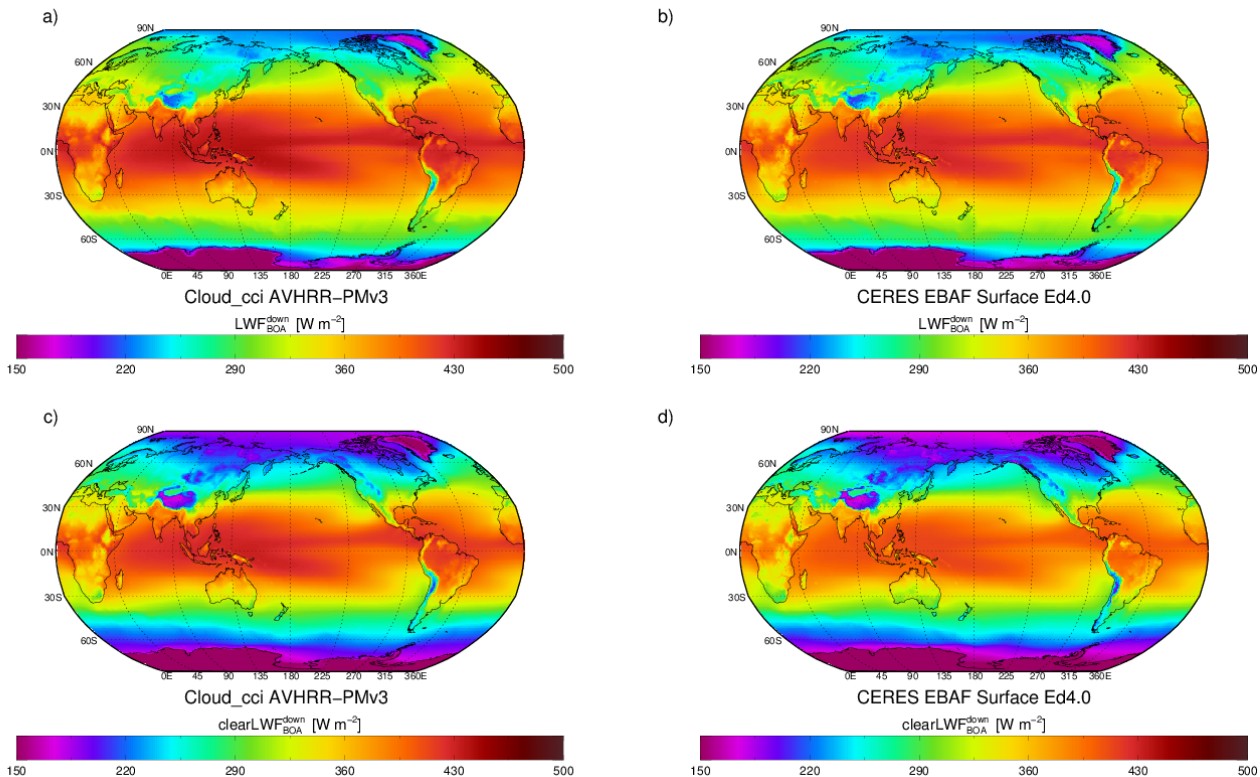

**Figure 6.** Multi-annual (2003-2016) mean downwelling longwave (LW) radiative fluxes at bottom-of-atmosphere (BOA) for all-sky conditions for Cloud_cci AVHRR-PMv3 (a) and CERES EBAF surface fluxes (b). Panels (c) and (d) show the same data but for clear-sky only.

**Table 8.** Multi-annual (2003-2016), latitude-weighted, global mean downwelling broadband shortwave and longwave fluxes (SWF, LWF) at bottom-of-atmosphere (BOA) inferred from Cloud_cci AVHRR-PMv3 dataset for all-sky and clear-sky (clear) conditions. The values are compared to equivalents inferred from Clouds and the Earth's Radiant Energy System (CERES) Energy Balanced and Filled (EBAF) surface fluxes. All values are given in W m$^{-2}$. In addition, differences and relative differences (Cloud_cci-CERES) for all fluxes are reported.

|  | $\mathrm{SWF}_{\mathrm{BOA}}^{\mathrm{down}}$ | $\mathrm{clearSWF}_{\mathrm{BOA}}^{\mathrm{down}}$ | $\mathrm{LWF}_{\mathrm{BOA}}^{\mathrm{down}}$ | $\mathrm{clearLWF}_{\mathrm{BOA}}^{\mathrm{down}}$ |
|---|---|---|---|---|
| Cloud_cci AVHRR-PMv3 [W m$^{-2}$] | 188.2 | 246.1 | 353.4 | 325.0 |
| CERES EBAF Ed.4.0 [W m$^{-2}$] | 187.3 | 243.9 | 345.4 | 314.6 |
| difference [W m$^{-2}$] | +0.9 | +2.2 | +8.0 | +10.4 |
| rel. difference | +0.5 % | +0.9 % | +2.3 % | +3.3 % |





**Table 9.** As Table 8 but for BOA upwelling broadband fluxes.

|  | $\text{SWF}_{\text{BOA}}^{\text{up}}$ | $\text{clearSWF}_{\text{BOA}}^{\text{up}}$ | $\text{LWF}_{\text{BOA}}^{\text{up}}$ | $\text{clearLWF}_{\text{BOA}}^{\text{up}}$ |
|---|---|---|---|---|
| Cloud_cci AVHRR-PMv3 [W m$^{-2}$] | 20.6 | 25.1 | 400.3 | 400.3 |
| CERES EBAF Ed.4.0 [W m$^{-2}$] | 23.3 | 29.7 | 398.8 | 398.1 |
| difference [W m$^{-2}$] | -2.7 | -4.6 | +1.5 | +2.2 |
| rel. difference | -11.6 % | -15.5 % | +0.4 % | +0.5 % |

### 3.3.2 TOA radiative fluxes

The Cloud_cci TOA radiative fluxes $\text{SWF}_{\text{TOA}}^{\text{up}}$ and $\text{LWF}_{\text{TOA}}^{\text{up}}$ were compared against the CERES EBAF TOA Edition-4.0 data (Loeb et al., 2018). As for the BOA fluxes the comparison includes multi-annual mean maps for the period 2003-2016. Figure 7 shows the maps for SWF for all-sky and clear-sky (clearSWF$_{\text{TOA}}^{\text{up}}$) conditions. Cloud_cci global patterns are very similar to those

of the CERES products. High $\text{SWF}_{\text{TOA}}^{\text{up}}$ values are found in regions with high surface albedo, e.g. deserts and polar regions, or with high cloud frequency, e.g. in mid-latitude storm track regions in both hemispheres, in the inner-tropical convergence zone and in regions with persistent marine stratocumulus clouds. Most prominent regions with low $\text{SWF}_{\text{TOA}}^{\text{up}}$ values are the subtropical subsidence regions (low cloud frequency) over the ocean (low surface albedo). It can also be seen that Cloud_cci provides slightly higher values in regions with high SWF (mainly land) and slightly lower values in regions with low SWF

(mainly ocean), compared to CERES. The comparisons of the clear-sky fluxes give very similar results. The global mean values given in Table 10 reveal differences of 2.9 and -3.3 W m$^{-2}$ for all-sky and clear-sky fluxes, respectively. The smaller values in Cloud_cci clear-sky is partly explained by the differences already found for $\text{SWF}_{\text{BOA}}^{\text{up}}$ (see previous section).

Figure 8 shows the results of an equivalent analysis for upwelling LWF at TOA. The global Cloud_cci patterns are again in very good agreement to CERES. High $\text{LWF}_{\text{TOA}}^{\text{up}}$ are mainly found in tropical and subtropical regions (high surface temperature)

with low cloud frequency or in regions with mainly low level clouds (marine stratocumulus or trade cumulus regions), where the cloud top temperatures are relatively warm. On the contrary, $\text{LWF}_{\text{TOA}}^{\text{up}}$ is low in regions with cold surfaces (e.g. polar regions) and regions with a high frequency of cold clouds. Cloud_cci clearLWF$_{\text{TOA}}^{\text{up}}$ are dominated by surface temperatures, thus decreasing towards higher latitudes; generally showing a very good agreement to CERES. The values given in (Table 10) reveal slightly smaller global mean values for Cloud_cci compared to CERES. This difference is almost doubled when considering

clear-sky fluxes, which is likely due to different sampling approaches. While for Cloud_cci all conditions are included (but removing the clouds when existent), CERES clear-sky TOA fluxes are determined by including clear-sky conditions only. This has the potential to bias TOA longwave fluxes high as clear-sky conditions have less water vapour (Sohn et al., 2010).



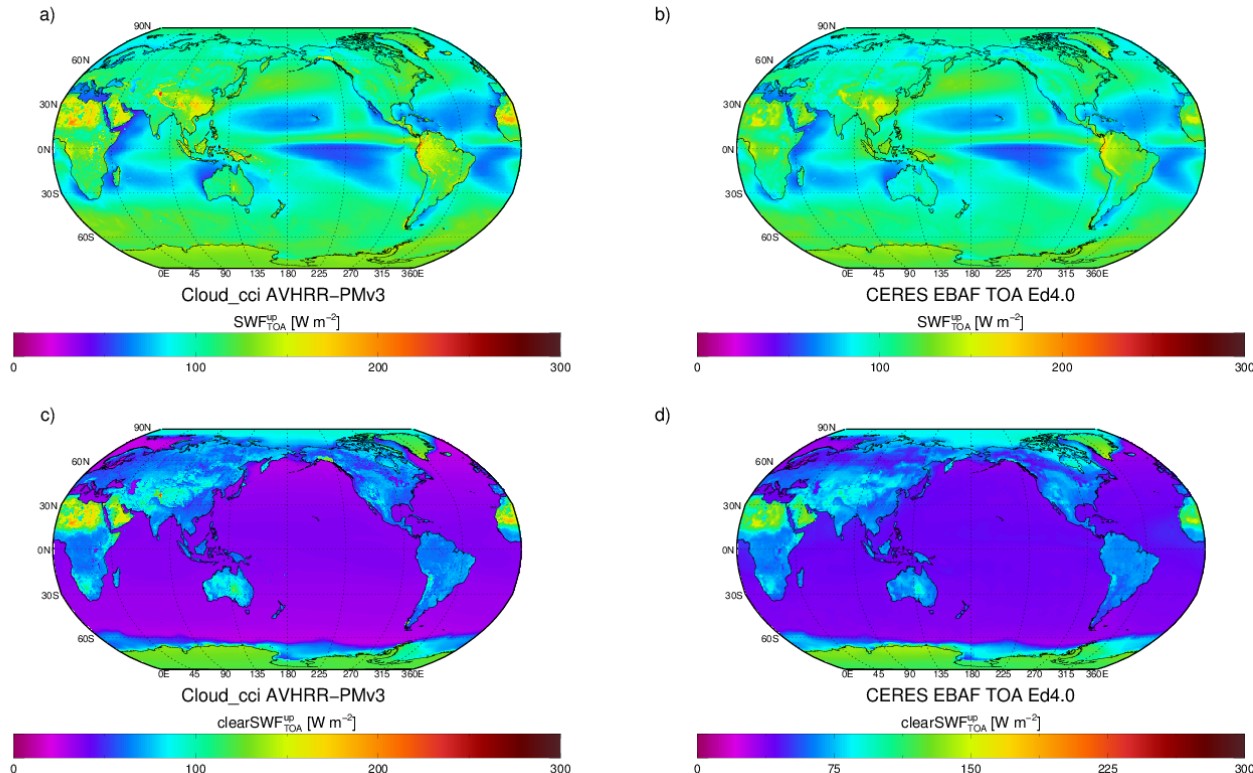

**Figure 7.** Multi-annual (2003-2016) mean top-of-atmosphere (TOA) upwelling shortwave (SW) radiative fluxes for all-sky conditions for Cloud_cci AVHRR-PMv3 (a) and CERES EBAF TOA edition 4.0 (b). Panels (c) and (d) show the same data but for clear-sky conditions.

**Table 10.** Multi-annual (2003-2016), latitude-weighted, global mean broadband fluxes at the top-of-atmosphere (TOA) inferred from Cloud_cci AVHRR-PMv3 dataset for all-sky and clear-sky (clear) conditions. The values are compared to equivalents inferred from Clouds and the Earth's Radiant Energy System (CERES) Energy Balanced and Filled (EBAF) TOA Edition-4.0. All values are given in W m$^{-2}$. In addition, differences and relative differences (Cloud_cci-CERES) of all fluxes are reported.

| | $SWF_{TOA}^{down}$ | $SWF_{TOA}^{up}$ | $clearSWF_{TOA}^{up}$ | $LWF_{TOA}^{up}$ | $clearLWF_{TOA}^{up}$ |
|---|---|---|---|---|---|
| Cloud_cci AVHRR-PMv3 [W m$^{-2}$] | 340.5 | 101.9 | 50.0 | 236.4 | 261.1 |
| CERES EBAF Ed.4.0 [W m$^{-2}$] | 340.3 | 99.0 | 53.3 | 240.3 | 268.3 |
| difference [W m$^{-2}$] | +0.2 | +2.9 | -3.3 | -3.9 | -7.2 |
| rel. difference | +0.1 % | +2.9 % | -6.2 % | -1.6 % | -2.7 % |

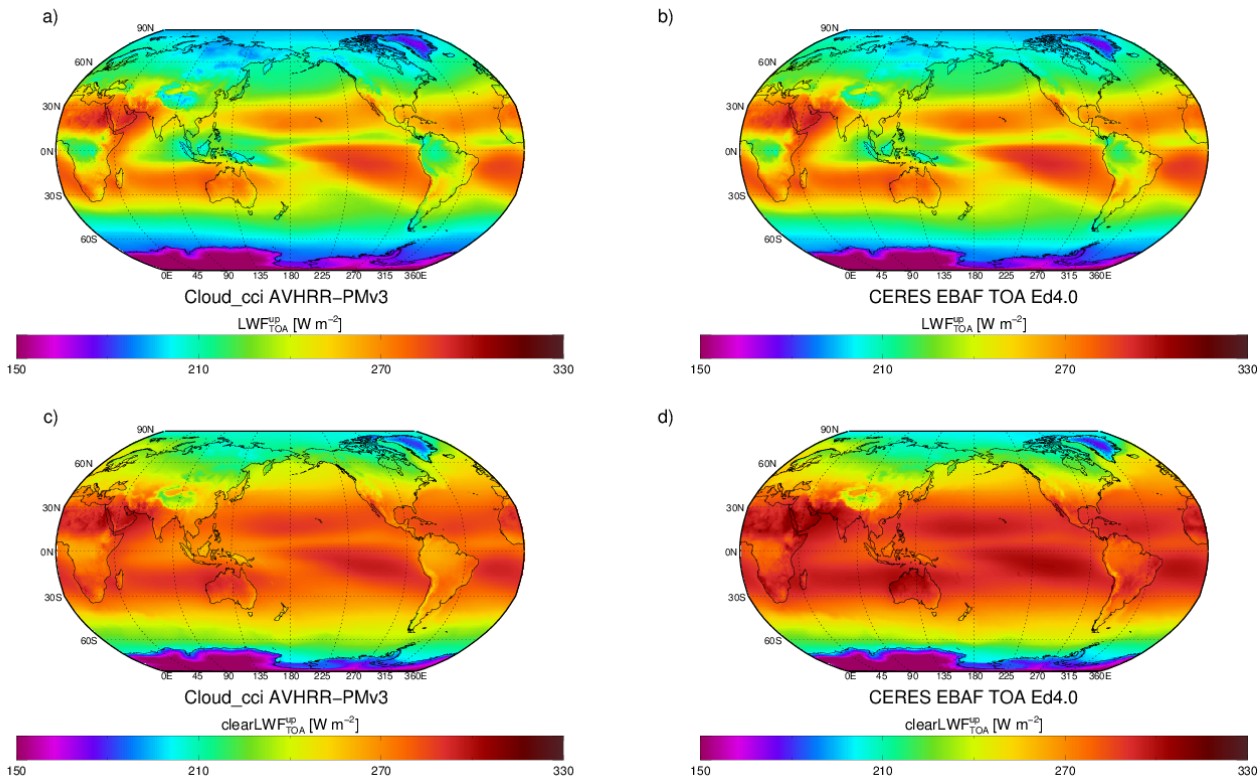

**Figure 8.** As Figure 7 but for TOA upwelling longwave (LW) fluxes.

## 4  Summary

As described in this paper, version 3 of the Cloud_cci AVHRR-PM dataset has been generated (and linked to a DOI, (Stengel et al., 2019)). In addition to clouds properties, this new version extends the product portfolio by BOA and TOA broadband radiative fluxes and covers the time period 1982 to 2016.

5      The cloud properties in v3 are superior to v2 in many aspects. This is demonstrated by analyses of global validation results against CALIOP (used for cloud detection, cloud phase, cloud top height), AMSR-E (used for liquid water path) and DARDAR (combined CALIOP and CPR information, used for ice water path). Heidke Skill Scores have increased from 0.64 to 0.68 for cloud detection and from 0.56 to 0.62 for cloud phase assignment. The scores are generally sensitive to whether or not thin clouds are included in the statistical comparisons. The improvements for cloud detection and phase determination in v3 remain
10   conclusive also for scenarios in which very thin clouds are excluded. The validation scores for cloud top height assignment remain nearly identical for liquid clouds, whereas for ice clouds lower standard deviations (2.36 km vs. 2.51 km) but larger negative biases (-3.54 km vs. -2.59 km) are found in v3. Similar results are found for scenarios in which the reference height is taken from below the geometrical top with penetrations optical depths of 0.15 and 1.0. Validation results for liquid water path





show a slight reduction in standard deviation for v3 from 27.1 g m$^{-2}$ to 26.4 g m$^{-2}$, accompanied by a slight increase in bias from -1.9 g m$^{-2}$ to -3.2 g m$^{-2}$. Correlations remain unchanged at 0.65. Ice water path validation shows reductions of standard deviations for v3 from 1299.8 g m$^{-2}$ to 900.9g m$^{-2}$ compared to v2 (reduction by 30 %). While the clearly increased correlation coefficient emphasises the improvement in v3 as well, the biases are somewhat larger in v3 compared to v2.

A new contribution to version 3 was the addition of top and bottom-of-atmosphere broadband radiative fluxes. Validation of v3 monthly mean downwelling radiative fluxes at BOA against BSRN stations reveals a very good agreement with low standard deviations of 13.8 W m$^{-2}$ for shortwave and 11.5 W m$^{-2}$ for longwave fluxes and correlation coefficients above 0.98 for both. While the bias for shortwave is small (1.9 W m$^{-2}$), a somewhat larger positive bias is found for longwave (7.6 W m$^{-2}$) which is mainly driven by moderate overestimations of larger flux values, but can potentially also partly be due to overestimations in
the reference (BSRN).

Comparisons of v3 multi-annual mean values of upwelling and downwelling fluxes at BOA and TOA with CERES additionally emphasises the good quality of the Cloud_cci radiative fluxes in terms of relative spatial pattern and absolute values. Concerning the latter, global mean values of Cloud_cci agree with CERES within 3.3 % for downwelling fluxes at BOA with larger deviations found for longwave fluxes. In contrast, Cloud_cci upwelling longwave fluxes at BOA agree very well with
CERES (below 0.5 %), upwelling shortwave fluxes at BOA show deviations up to about 15 %, although the absolute differences are only 4.6 W m$^{-2}$ at maximum. It however remains uncertain to which extent uncertainties in CERES products contribute to these deviations.

In contrast to the BOA, CERES products for TOA fluxes are mainly based on observational information, thus provide an excellent reference for validation. For all-sky fluxes Cloud_cci agrees to CERES within 3% for global mean values. The
differences are increased when considering clear-sky fluxes. It is likely that the different approaches to estimate the mean clear-sky fluxes in Cloud_cci (including all conditions, but removing the clouds) and CERES (including only cloud-free conditions) contribute considerably to these differences.

In summary, Cloud_cci AVHRR-PMv3 represents a dataset of consistent cloud properties and radiative fluxes, which in many aspects is superior to the precursor version v2 as data quality was improved, the product portfolio extended and the covered
time period prolonged. Cloud_cci AVHRR-PMv3 offers a large variety of applications including climatological analyses of cloud properties and radiative fluxes as well as their dependency to each other at time scales of several decades.

## 5   Data availability

For the presented dataset (Cloud_cci AVHRR-PMv3) a DOI has been issued: https://doi.org/10.5676/DWD/ESA_Cloud_cci/AVHRR-PM/V003 (Stengel et al., 2019), of which the landing page points to additional documentation and data download sites. A
parallel dataset based on AVHRR on board the NOAA and EUMETSAT morning satellites exists (AVHRR-AMv3), for which a DOI has been issued as well: https://doi.org/10.5676/DWD/ESA_Cloud_cci/AVHRR-AM/V003. The CC4CL retrieval system used to produce the data is version controlled and accessible at github: https://github.com/ORAC-CC/orac/wiki. The LUT



creation code is available at https://github.com/ORAC-CC/create_orac_lut. Both are licensed under the GNU General Public License (GPL) version 3.

## Appendix A: AVHRR measurement data

The AVHRR measurement record used as basis for the presented cloud climatology spans the AVHRR/2 and AVHRR/3 sensor
5  generations on board NOAA-7, NOAA-9, NOAA-11, NOAA-14, NOAA-16, NOAA-18 and NOAA-19. Based on the original AVHRR measurements (Local Area Coverage) with 1 km spatial resolution and sampling distance, the Global Area Coverage (GAC) data is globally available, but with reduced spatial resolution and sampling distance. Only every fourth scanline is used and within one scanline four neighbouring pixels are averaged. The AVHRR sensor has an on-board black-body calibration mechanism for its infrared channels. No attempt is made to further recalibrate these measurements. For the visible channels,
10 no calibration is performed on board AVHRR. A recalibration procedure for these channels was applied as a preparatory step based on Devasthale et al. (2017) with further application aspects reported in Schlundt et al. (2017).

## Appendix B: Measurement input to the ANNs and the thresholds applied posterior

**Table B1.** Measurement input to the trained artificial neural network for cloud detection ($ANN_{mask}$), used for different illumination conditions: daytime, twilight and night-time. The subscript in the table's headline corresponds to the approximate central wavelengths of the channels: 0.6μm, 0.8μm, 1.6μm, 3.7μm, 10.8μm, 12.0μm. In addition to the measurement input, all ANNs require surface temperature, a snow-ice flag and a land-sea flag as input. R=reflectance, BT=brightness temperature

| $ANN_{mask}$ | $R_{0.6}$ | $R_{0.8}$ | $R_{1.6}$ | $R_{3.7}$ | $BT_{3.7}$ | $BT_{10.8}$ | $BT_{12.0}$ | $BT_{10.8}-BT_{12.0}$ | $BT_{10.8}-BT_{3.7}$ |
|---|---|---|---|---|---|---|---|---|---|
| Day | ✓ | ✓ | - | ✓ | - | ✓ | ✓ | ✓ | - |
| Twilight | - | - | - | - | ✓ | ✓ | ✓ | ✓ | ✓ |
| Night | - | - | - | - | ✓ | ✓ | ✓ | ✓ | ✓ |





**Table B2.** Empirical thresholds used to convert the output of the cloud mask ANNs to a binary cloud mask. Thresholds depend on illumination conditions and surface type.

| Illumination | Surface type | Threshold |
|---|---|---|
| Day | Sea ice | 0.4 |
| Day | Land ice | 0.3 |
| Day | Sea | 0.25 |
| Day | Land | 0.3 |
| Night | Sea ice | 0.45 |
| Night | Land ice | 0.35 |
| Night | Sea | 0.25 |
| Night | Land | 0.3 |
| Twilight | Sea ice | 0.5 |
| Twilight | Land ice | 0.35 |
| Twilight | Sea | 0.35 |
| Twilight | Land | 0.45 |

**Table B3.** Measurement input to the trained artificial neural network for cloud phase determination ($\text{ANN}_{\text{phase}}$), used for different illumination conditions: daytime, twilight and night-time. The subscript in the table's headline corresponds to the approximate central wavelength of the channels: 0.6 μm, 0.8 μm, 1.6 μm, 3.7 μm, 10.8 μm, 12.0 μm. In addition to the measurement input, all ANNs require a surface type flag containing the values 0:sea,1:land,2:desert,3:sea-ice,4:snow.

| $\text{ANN}_{\text{phase}}$ | $R_{0.6}$ | $R_{0.8}$ | $R_{1.6}$ | $R_{3.7}$ | $BT_{3.7}$ | $BT_{10.8}$ | $BT_{12.0}$ | $BT_{10.8}$-$BT_{12.0}$ | $BT_{10.8}$-$BT_{3.7}$ |
|---|---|---|---|---|---|---|---|---|---|
| Day | ✓ | ✓ | - | ✓ | - | ✓ | ✓ | ✓ | - |
| Twilight | - | - | - | - | ✓ | ✓ | ✓ | ✓ | ✓ |
| Night | - | - | - | - | ✓ | ✓ | ✓ | ✓ | ✓ |





**Table B4.** Empirical thresholds used to convert the output of the cloud phase ANNs to a binary cloud phase. Thresholds depend on illumination conditions and surface types.

| Illumination | Surface type | Threshold |
|:---:|:---:|:---:|
| Day | Sea ice | 0.5 |
| Day | Land ice | 0.7 |
| Day | Sea | 0.55 |
| Day | Land | 0.7 |
| Night | Sea ice | 0.7 |
| Night | Land ice | 0.6 |
| Night | Sea | 0.5 |
| Night | Land | 0.65 |
| Twilight | Sea ice | 0.7 |
| Twilight | Land ice | 0.9 |
| Twilight | Sea | 0.65 |
| Twilight | Land | 0.50 |

## Appendix C: Spectral band adjustment (SBA)

As the cloud detection and cloud phase determination were developed and fine-tuned primarily based on NOAA-19 AVHRR, adjustment factors (slope and offset) were inferred to make all considered AVHRR sensors mimic NOAA-19 AVHRR. The SBAs were inferred from a set of SCIAMACHY and IASI orbits with both of these sensors providing hyperspectral measurements throughout the visible (SCIAMACHY) and infrared (IASI) part of the spectrum, respectively. Using the spectral response functions (SRF) of AVHRR channels 0.6 μm, 0.8 μm, 10.8 μm and 12.0 μm the SCIAMACHY and IASI measurements were convolved to mimic synthetic AVHRR measurements in each footprint of the considered SCIAMACHY and IASI orbits. Using this procedure for all AVHRR sensors (the AVHRR SRFs differ among the individual satellites) and collecting the synthetic AVHRR measurements in all considered footprints of SCIAMACHY and IASI, a database was composed allowing for linearly fitting all AVHRR sensors to AVHRR onboard NOAA-19. This SBA is applied prior to the application of the cloud detection and cloud phase procedures. No attempt is made to adjust channels 1.6 μm and 3.7 μm as the SCIAMACHY and IASI spectra do not cover the full AVHRR SRF of these channels. All inferred SBAs are given in Table C1. In v2 of the datasets, no SBAs were applied among the AVHRR sensors. As the OE retrieval makes direct use of the SRF of the individual AVHRR sensors, the application of the SBA is not required for the OE retrieval.





**Table C1.** Linear regression coefficients (slope and offset) applied as spectral band adjustment to either measured reflectances (Rs) or brightness temperature (TBs) of all used AVHRR channels (Ch) and all used sensors to mimic NOAA-19 AVHRR. The subscript in the table's headline corresponds to the approximate central wavelengths of the channels: 0.6μm, 0.8μm, 1.6 μm, 3.7 μm, 10.8 μm, 12.0 μm. Reflectances in channels 0.6 μm, 0.8 μm and 1.6 μm are generally not used in twilight and night conditions.

| | $R_{0.6}$ slope \| offset | $R_{0.8}$ slope \| offset | $R_{1.6}$ slope \| offset | $BT_{3.7}$ slope \| offset | $BT_{10.8}$ slope \| offset | $BT_{12.0}$ slope \| offset |
|---|---|---|---|---|---|---|
| | | | day | | | |
| NOAA-7 | 1.009 \| -0.036 | 1.007 \| -0.007 | 1.000 \| 0.000 | 1.000 \| 0.000 | 1.000 \| -0.198 | 0.991 \| 1.991 |
| NOAA-9 | 1.009 \| -0.013 | 1.006 \| 0.011 | 1.000 \| 0.000 | 1.000 \| 0.000 | 1.000 \| -0.215 | 0.988 \| 2.770 |
| NOAA-11 | 1.009 \| -0.010 | 1.005 \| -0.012 | 1.000 \| 0.000 | 1.000 \| 0.000 | 1.000 \| -0.170 | 0.989 \| 2.443 |
| NOAA-14 | 1.008 \| 0.016 | 1.011 \| -0.026 | 1.000 \| 0.000 | 1.000 \| 0.000 | 1.001 \| -0.446 | 0.995 \| 1.081 |
| NOAA-16 | 1.006 \| -0.039 | 1.009 \| 0.057 | 1.000 \| 0.000 | 1.000 \| 0.000 | 1.000 \| -0.095 | 0.997 \| 0.561 |
| NOAA-18 | 1.002 \| -0.013 | 1.015 \| 0.066 | 1.000 \| 0.000 | 1.000 \| 0.000 | 1.000 \| -0.214 | 0.997 \| 0.626 |
| NOAA-19 | 1.000 \| 0.000 | 1.000 \| -0.000 | 1.000 \| 0.000 | 1.000 \| 0.000 | 1.000 \| -0.000 | 1.000 \| 0.000 |
| | | | twilight & night | | | |
| NOAA-7 | - | - | - | 1.000 \| 0.000 | 1.000 \| -0.194 | 0.992 \| 1.786 |
| NOAA-9 | - | - | - | 1.000 \| 0.000 | 1.000 \| -0.243 | 0.989 \| 2.500 |
| NOAA-11 | - | - | - | 1.000 \| 0.000 | 1.000 \| -0.178 | 0.990 \| 2.184 |
| NOAA-14 | - | - | - | 1.000 \| 0.000 | 1.001 \| -0.427 | 0.996 \| 0.945 |
| NOAA-16 | - | - | - | 1.000 \| 0.000 | 1.000 \| 0.022 | 0.997 \| 0.511 |
| NOAA-18 | - | - | - | 1.000 \| 0.000 | 1.000 \| -0.209 | 0.997 \| 0.542 |
| NOAA-19 | - | - | - | 1.000 \| 0.000 | 1.000 \| 0.000 | 1.000 \| 0.000 |

*Author contributions.* MS coordinated the generation of the presented dataset, contributed to key developments and drafted the manuscript. SF prepared the AVHRR measurement record used as input. MS, SS and OS developed the cloud detection and phase determination and implemented the processing system used for the generation of the multi-decadal data set. CP and GM further developed the optimal estimation system. MC implemented the BUGSrad scheme used for the calculation of the radiative flux properties. MS, SS, BW and DP evaluated the
5    data. All authors contributed to finalizing the manuscript.

*Competing interests.* The authors declare that no competing interests are present.



*Acknowledgements.* This work was supported by the European Space Agency (ESA) through the Cloud_cci project (contract No.: 4000109870/13/I-NB). The availability of the AVHRR GAC measurement record through the NOAA CLASS archive (https://www.class.noaa.gov) and the University of Wisconsin is much appreciated, with the latter also kindly providing corresponding intercalibration coefficients for the visible and near-infrared channels of AVHRR. Furthermore, the authors would like to thank Luca Lelli (University of Bremen) for providing SCIAMACHY data used to determine the spectral band adjustment.



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
