# Peer review of "Cloud\_cci AVHRR-PM dataset version 3: 35 year climatology of global cloud and radiation properties"

_Earth System Science Data, 2019_

## Referee Comment (RC1) · Anonymous Referee #1 · 8 Aug 2019

This article describes the third version of the Cloud Climate Change Initiative AVHRR-PM dataset; a 35 year climatology based on measurements in 5 spectral bands from the Advanced Very High Resolution Radiometer (AVHRR) instruments on board several polar orbiting satellites. This dataset includes both cloud properties retrieved from the AVHRR measurements and surface and top of atmosphere irradiances calculated using these retrieved cloud properties. The article describes changes since the previous version of the dataset and presents some evaluation of the dataset and comparisons with the previous version.

This article is generally very well written. The description of the cloud retrieval algorithm

is quite brief, but this is appropriate as the previous version of the algorithm is described in detail in a previous publication (Stengel et al. 2017). The description of the radiative transfer calculations is also quite brief and in my opinion more detail is required here (see comments below for suggestions). The dataset was accessed through the given identifier and appeared to be complete and consistent with the description in the article.

**Recommendation**

Accept pending minor revisions.

**Minor Comments**

1. Why does this article focus specifically on AVHRR-pm, as opposed to describing datasets for multiple instruments as in Stengel et al (2017)? At the very least, it seems odd not to include the AVHRR-am dataset in the scope of this article. On similar lines, would it be possible to produce a product combining AVHRR-am and AVHRR-pm measurements? Presumably the additional sampling of the diurnal cycle would lead to smaller errors in the radiation diurnal cycle corrections.

2. It's very difficult to see any differences in most of Figs. 1,2,5,6,7,8. I would consider including difference plots, either instead of the v2/CERES images, or as an additional row/column.

3. Is there any attempt to account for changes in the surface albedo with the angle of incident light in the SW radiative transfer calculations (e.g. Wang et al 2007)? Perhaps this could explain some of the differences between the CERES and AVHRR-pm surface SW upwelling irradiances?

4. With regards to the LW diurnal cycle correction factors, are separate factors derived for clear and cloudy scenes?

5. If I understand the radiative transfer model correctly, it requires the cloud to be split into layers. If this is the case, how do you determine how many layers to include cloud in (i.e. where is the cloud base?). I would expect this to have a reasonably large impact on the calculated surface LW downwelling irradiance.

6. Page 15, Line 11 -13. I don't follow the argument that "the larger standard deviations... is primarily related to variances in surface albedo and cloud cover which tend to have significant annual cycles". Relating the larger s.d. to the surface albedo variance makes sense, but I don't understand why the cloud cover variance will lead to a larger s.d. as it also affects the downwelling SW irradiance.

7. For TOA radiation, clear-sky differences between CERES and AVHRR-pm are attributed to sampling differences. Presumably this is relatively easy to test by calculating a CERES-like value from the AVHRR-pm product?

8. To further demonstrate the usefulness of the radiation products, it would be good to see some further comparisons with other datasets, such as the ERA-Interim reanalysis, or the GEWEX radiation budget data. Perhaps you could add a couple of extra lines in table 8 to show mean values for other products?

9. I really appreciate the effort undertaken to provide useful and accurate uncertainty estimates for the cloud variables. It would be very helpful to have some estimate of uncertainty in the computed radiation variables too. This could be based on further radiative transfer calculations using different cloud inputs to represent the uncertainty in the input cloud profiles, though this may be time consuming. Alternatively, a simple quality variable to indicate when the radiation calculation is uncertain due to larger uncertainty in the input cloud profiles could potentially be quite helpful.

10. Looking at the daily data, there appear to be some artifacts in the retrieved cloud water path at the edges of the swaths for the descending overpasses (e.g. for the 1 June 2016 data). These do seem to correspond with very large uncertainty estimates. In such cases, where the uncertainty is much larger than the retrieved value I wonder whether it would be better to replace the retrieved value with a missing data value? In particular, I have concerns about these retrieved values undergoing further processing (e.g. passed to radiation calculations, or used in monthly mean/histogram products) and the information about the large uncertainty associated with the particular retrieval being lost.

**References**

- Stengel, M., Stapelberg, S., Sus, O., Schlundt, C., Poulsen, C., Thomas, G., Christensen, M., Carbajal Henken, C., Preusker, R., Fischer, J., Devasthale, A., Willén, U., Karlsson, K.-G., McGarragh, G. R., Proud, S., Povey, A. C., Grainger, R. G., Meirink, J. F., Feofilov, A., Bennartz, R., Bojanowski, J. S., and Hollmann, R.: Cloud property datasets retrieved from AVHRR, MODIS, AATSR and MERIS in the framework of the Cloud_cci project, Earth Syst. Sci. Data, 9, 881-904, https://doi.org/10.5194/essd-9-881-2017, 2017.

- Wang, Z., Zeng, X., and Barlage, M. ( 2007), Moderate Resolution Imaging Spectroradiometer bidirectional reflectance distribution function–based albedo parameterization for weather and climate models, J. Geophys. Res., 112, D02103, doi:10.1029/2005JD006736.

---

## Referee Comment (RC2) · Anonymous Referee #2 · 12 Aug 2019

Summary of paper: The authors describe a data set of global cloud properties based on AVHRR observations, available since 1982. The data set is an update of version 2, with the main changes the use of artificial neural networks for cloud mask and cloud phase detection, and additional cloud radiative properties. Both versions 2 and 3 are evaluated against the best available retrievals from other satellite and ground-based products. Standard verification metrics indicate overall improvement in most cloud properties, with some deterioration in ice cloud top height. The cloud radiative properties compare well against CERES observations.

Review: This paper is generally very well written. It is mostly complete and useful

information for anyone wishing to use this data set. Comments mostly concern some further clarification sought and perhaps slightly different presentation of the differences between version 2 and 3. The DOI links to a suitably presented web page describing the data. Overall, the recommendation is to accept this paper with minor corrections.

Minor comment:

Statistical significance. It is not immediately clear whether any of the differences in skill between v2 and v3 reported in the tables are statistically significant, although the large sample suggests these are. However, it should be possible and it would help the reader if the maps in figures 1, 2, 5, 6, 7, and 8 could include (i) difference/bias maps and (ii) stippling/hashing/shading for statistically significant differences. Most of these maps are visually similar and might hide key differences due to the colour scale used. A different way of presenting the various data sets, including additional maps of bias and statistically significant differences, would help inform the reader how the new data set compares against existing data sets.

Line-by-line comments:

p2. l28-30. This sentence is difficult to read, especially the first part.

p2. l33. "limitations". Limitations to do what?

p3. l3-7. Please provide references for the WCRP GEWEX data and the ISCCP DX data.

p3. l10-14. "based on the rationale above". It is clear why these data are required, compared to the MODIS/CERES and GEWEX data sets. However, what are the other data sets based on AVHRR lacking (PATMOS-x, CLARA-A2, Cloud-cci) that this paper will address with Cloud cci 3? A sentence on p2, line 19-21 would help clarify the shortcomings of those existing data sets.

p3. l20-31. This paragraph pre-empts the findings ("superior") and methodology. The relevant information is better placed in section 2.1.

p4. l2. Please add that table 1 contains all abbreviations used throughout the text. CER had not been introduced in the main text prior to p6 and it took a moment to figure out its meaning.

p4. l16. "much larger set". How do the two sets of training data compare? Did both v2 and v3 use CALIOP, but v2 just used fewer overpasses?

p6. l16. A "lower" CTP mean is not explained by more very low-level clouds, which have higher CTP. It appears that over the West Pacific and Maritime Continent, mean CTP has generally increased, which could be due to detection of more low-level clouds. Please re-consider this statement.

p7. l1. Regarding the validation, did the authors consider performing the validation separately for daytime and nighttime observations? The algorithms use different channels and the authors consider nighttime COT and CER "experimental". It would be useful to understand the algorithm performance for different times of the day.

p7. l3. Please, briefly explain how the collocation is carried out. In particular, what is the impact of the temporal mismatch between CALIOP and AVHRR? And what is the impact of the mismatch in footprint?

p7. l22-23. Why would improved identification of liquid clouds lead to reduced POD for ice clouds? This suggests that some ice clouds are now erroneously identified as liquid. Does that mean there are more "false alarms" in terms of liquid cloud detection?

p8. l5-15. It would be helpful to consider the results from Tables 4, 5, and 6 through a visual comparison, as done in Figure 4. A scatter plot (or 2D histogram) of CTH, LWP, and IWP comparing the data set with the "truth" could help identify where biases are most likely to occur. For instance, the CTH bias of ice cloud could be mostly due to the highest clouds, even at high COT, as these might have a region of low extinction coefficient near cloud top, that would lead to higher CTH in CALIOP. A scatter plot could show this clearly. Similarly, LWP and IWP are highly skewed variables and the metrics

presented could be affected by a few outliers. A scatter plot or 2D histogram (perhaps shown on a logarithmic scale) could indicate whether LWP and IWP estimates are typically good, or whether there is a consistent bias across cloud types of all LWP and IWP values.

p9. p10. p11. Please rename standard deviation to "root mean squared error", which is presumably what is reported.

---

## Author Comment (AC1) · 1 Nov 2019

**General comments:**

This article describes the third version of the Cloud Climate Change Initiative AVHRR-PM dataset; a 35 year climatology based on measurements in 5 spectral bands from the Advanced Very High Resolution Radiometer (AVHRR) instruments on board several polar orbiting satellites. This dataset includes both cloud properties retrieved from the AVHRR measurements and surface and top of atmosphere irradiances calculated using these retrieved cloud properties. The article describes changes since the previous version of the dataset and presents some evaluation of the dataset and comparisons with the previous version.

This article is generally very well written. The description of the cloud retrieval algorithm is quite brief, but this is appropriate as the previous version of the algorithm is described in detail in a previous publication (Stengel et al. 2017). The description of the radiative transfer calculations is also quite brief and in my opinion more detail is required here (see comments below for suggestions). The dataset was accessed through the given identifier and appeared to be complete and consistent with the description in the article.

**Recommendation:**

Accept pending minor revisions.

**Minor Comments:**

*Referee comment:*

1. Why does this article focus specifically on AVHRR-pm, as opposed to describing datasets for multiple instruments as in Stengel et al (2017)? At the very least, it seems odd not to include the AVHRR-am dataset in the scope of this article. On similar lines, would it be possible to produce a product combining AVHRR-am and AVHRR-pm measurements? Presumably the additional sampling of the diurnal cycle would lead to smaller errors in the radiation diurnal cycle corrections.

*Author's response:*

The Stengel et al. (2017) paper aimed at introducing all available version 2 datasets generated in the framework of the Cloud_cci project. Only a subset of those (AVHRR-AM, AVHRR-PM, ATSR2-AATSR) were reprocessed building on new developments leading to corresponding version 3 datasets. ATSR2-AATSR version 3 data will be introduced in a separate paper, which is soon to be submitted. For AVHRR we decided to put the focus more or less entirely on AVHRR-PM as this dataset is longer, more stable and of higher quality than AVHRR-AM. On the other hand, for the period covered by the AM satellites NOAA-17 and METOP-A (NOAA-12 and NOAA-15 are very difficult to handle due to their twilight orbit), the AVHRR-AM datasets is of similar quality as AVHRR-PM and does indeed provided the possibility for combining AVHRR-AM and AVHRR-PM to reduced sampling problems, although only for the years 2002 and beyond. Another difficulty is the availability of the 1.6mic channel instead of the 3.7mic channel as available on nearly all PM satellites. In the data availability section, the existence of the AVHRR-AMv3 is reflected. We will add that there is a potential to combine with AVHRR-PMv3, but also mentioning the difficulties for NOAA-12 and NOAA-15.

*Author's changes to the manuscript:*

We added the following sentence to the data availability section: "The AVHRR-AMv3 dataset provides the feasibility to be combined with AVHRR-PMv3 to increase sampling frequency. However, for the period of NOAA-12 and NOAA-15 the AVHRR-AMv3 dataset is of reduced quality due to the difficult twilight orbits of NOAA-12 and NOAA-15."

*Referee comment:*
2. It's very difficult to see any differences in most of Figs. 1,2,5,6,7,8. I would consider including difference plots, either instead of the v2/CERES images, or as an additional row/column.
*Author's response:*
We have included differences plots in Figures 1,2,5,6,7,8.
*Author's changes to the manuscript:*
We updated Figures 1,2,5,6,7,8.

*Referee comment:*
3. Is there any attempt to account for changes in the surface albedo with the angle of incident light in the SW radiative transfer calculations (e.g. Wang et al 2007)? Perhaps this could explain some of the differences between the CERES and AVHRR-pm surface SW upwelling irradiances?
*Author's response:*
Yes, for ocean surfaces we have built in an empirical method to adjust the surface albedo as a quadratic function of the angle of incident light. For land surfaces no adjustment is made. We will include a statement on this in the text.
*Author's changes to the manuscript:*
On page 13 line 2 we will change the sentence to "The diurnal cycle of SZA is then used to rescale the incoming and reflected solar radiation, adjust the surface albedo (using an empirical quadratic function of SZA) and the atmospheric path length for a given set of time stamps throughout the local day."

*Referee comment:*
4. With regards to the LW diurnal cycle correction factors, are separate factors derived for clear and cloudy scenes?
*Author's response:*
No, this is not the case, but probably something to consider for the future. Thanks. In this context we noticed that the LW diurnal cycle correction is only applied for land surfaces, which is not reflected in the manuscript yet. We will add this now.
*Author's changes to the manuscript:*
On page 13 line 7 we will modify the sentence to "For longwave radiation, a diurnal cycle correction is applied over land based on a cosine fit to an observed mean…."

*Referee comment:*
5. If I understand the radiative transfer model correctly, it requires the cloud to be split into layers. If this is the case, how do you determine how many layers to include cloud in (i.e. where is the cloud base?). I would expect this to have a reasonably large impact on the calculated surface LW downwelling irradiance.
*Author's response:*
We assume the radiative transfer in BUGSrad is meant which is employed to derive the broadband fluxes. Here we assume only one cloud layer with its top being place at the derived cloud top height. Using derived optical thickness and effective radius the geometrical thickness, thus the cloud base height is estimated. This is actually described in more detail in an ATBD, which we will include a reference to.
*Author's changes to the manuscript:*
On page 12 line 22 we will add the sentence "The reader is referred to ATBD (2019) for more details on the calculation of the broadband fluxes." Along with including the following reference :
ATBD – Algorithm Theoretical Baseline Document (ATBD) of CC4CL Broadband Radiative Flux Retrieval - ESA Cloud_cci, 2019, Issue 1, Rev. 1; 14/10/2019, available from http://www.esa-cloud-cci.org/?q=documentation, 2019.

*Referee comment:*
6. Page 15, Line 11 -13. I don't follow the argument that "the larger standard deviations...is primarily related to variances in surface albedo and cloud cover which tend to have significant annual cycles". Relating the larger s.d. to the surface albedo variance makes sense, but I don't understand why the cloud cover variance will lead to a larger s.d. as it also affects the downwelling SW irradiance.

*Author's response:*

Right. We will remove cloud cover.

*Author's changes to the manuscript:*

The revised version of that sentence will read "The larger standard deviations retrieved form the solar reflected radiation is primarily related to variances in surface albedo which tend to have significant annual cycles"

*Referee comment:*

7. For TOA radiation, clear-sky differences between CERES and AVHRR-pm are attributed to sampling differences. Presumably this is relatively easy to test by calculating a CERES-like value from the AVHRR-pm product?

*Author's response:*

Thank you for this suggestion. We performed a little experiment (covering 3 months) in which we emulated the CERES-like clear-sky sampling. And yes, when doing so the global mean TOA LW flux was increased by approx. 3W/m², thus the deviation to CERES was reduced. This emphasises that at least parts of the deviation can be explained by sampling. We will include the results of this experiment in the text.

*Author's changes to the manuscript:*

The following sentence was added: "This could be confirmed by a 3-months covering test run in which Cloud_cci LWF^up_toa was only averaged over clear-sky cases, which led to an increase by about 3~$\text{W m}^{\text{-2}}$ for the global mean value."

*Referee comment:*

8. To further demonstrate the usefulness of the radiation products, it would be good to see some further comparisons with other datasets, such as the ERA-Interim reanalysis, or the GEWEX radiation budget data. Perhaps you could add a couple of extra lines in table 8 to show mean values for other products?

*Author's response:*

Good suggestion. We will add value for ERA-Interim to table 8, and addition to tables 9 and 10. We prefer not to add the GEWEX SRB dataset as it does not fully cover the period 2003-2016 chosen for corresponding comparisons.

*Author's changes to the manuscript:*

We will add ERA-Interim values to Tables 8, 9 and 10.

*Referee comment:*

9. I really appreciate the effort undertaken to provide useful and accurate uncertainty estimates for the cloud variables. It would be very helpful to have some estimate of uncertainty in the computed radiation variables too. This could be based on further radiative transfer calculations using different cloud inputs to represent the uncertainty in the input cloud profiles, though this may be time consuming. Alternatively, a simple quality variable to indicate when the radiation calculation is uncertain due to larger uncertainty in the input cloud profiles could potentially be quite helpful.

*Author's response:*

Thank you for this suggestion. We have in indeed planned to provide uncertainty estimates for the radiation variables too. For the presented dataset version this was however not feasible due to time constrains, but we certainly have this on the to-do list for next versions. For the time being the radiation validation results presented do provide some guidance wrt. to certainty/uncertainty of the radiation products, although not on pixels/grid-cell level. As the determination of the radiation product heavily depend on the derived cloud properties and their uncertainty it is indeed wise to

inform the users of the data that the provided cloud property uncertainties give hints on the certainty/uncertainty in the radiation product already in the current dataset version. We will include a comment on this in the text.

*Author's changes to the manuscript:*
We added the following sentence at the end of Section 3.1 "In contrast to the cloud properties, the radiative fluxes in the presented dataset version are not accompanied by uncertainty estimates on pixel level. While the validation results presented below provide a general guidance to the quality of the radiative fluxes, user of the data are also encouraged to inspect the pixel-level uncertainties of the cloud properties as these are dominant input to the calculation of the fluxes."

*Referee comment:*
10. Looking at the daily data, there appear to be some artifacts in the retrieved cloud water path at the edges of the swaths for the descending overpasses (e.g. for the 1 June 2016 data). These do seem to correspond with very large uncertainty estimates. In such cases, where the uncertainty is much larger than the retrieved value I wonder whether it would be better to replace the retrieved value with a missing data value? In particular, I have concerns about these retrieved values undergoing further processing (e.g. passed to radiation calculations, or used in monthly mean/histogram products) and the information about the large uncertainty associated with the particular retrieval being lost.

*Author's response:*
Thank you for this observation and feedback. The descending cloud water path is based on the night-time retrievals of CER and COT, which are experimental products as listed in the manuscript. To emphasise this we will add a corresponding comment below Table 1. For monthly aggregations these night-time products are not considered. A quick inspection of the LW fluxes confirmed that this issue does not seemed to have a significant impact on the LW fluxes. SW fluxes are zero anyway during night time.

*Author's changes to the manuscript:*
Added a comment below Table 1.

---

## Author Comment (AC2) · 1 Nov 2019

**General comments:**

Summary of paper: The authors describe a data set of global cloud properties based on AVHRR observations, available since 1982. The data set is an update of version 2, with the main changes the use of artificial neural networks for cloud mask and cloud phase detection, and additional cloud radiative properties. Both versions 2 and 3 are evaluated against the best available retrievals from other satellite and ground-based products. Standard verification metrics indicate overall improvement in most cloud properties, with some deterioration in ice cloud top height. The cloud radiative properties compare well against CERES observations.

Review: This paper is generally very well written. It is mostly complete and useful information for anyone wishing to use this data set. Comments mostly concern some further clarification sought and perhaps slightly different presentation of the differences between version 2 and 3. The DOI links to a suitably presented web page describing the data. Overall, the recommendation is to accept this paper with minor corrections.

**Minor comment:**

*Referee comment:*
Statistical significance. It is not immediately clear whether any of the differences in skill between v2 and v3 reported in the tables are statistically significant, although the large sample suggests these are. However, it should be possible and it would help the reader if the maps in figures 1, 2, 5, 6, 7, and 8 could include (i) difference/bias maps and (ii) stippling/hashing/shading for statistically significant differences. Most of these maps are visually similar and might hide key differences due to the colour scale used. A different way of presenting the various data sets, including additional maps of bias and statistically significant differences, would help inform the reader how the new data set compares against existing data sets.

*Author's response:*
Thank you for this suggestion. We updated Figures 1,2,3,5,6,7 and 8 to include difference maps as suggested. Unfortunately, we have not always all information available to calculate and provide the statistical significance of the differences, which are for this reason not included.

*Author's changes to the manuscript:*
Updated Figures 1,2,3,5,6,7 and 8, now including difference maps.

**Line-by-line comments:**

*Referee comment:*
p2. l28-30. This sentence is difficult to read, especially the first part.

*Author's response:*
Thanks. We will rephrase that sentence.

*Author's changes to the manuscript:*
The rephrased sentence will read "For the MODIS cloud record however there is the potential to be combined with high quality TOA radiation measurements made by the Clouds and the Earth's Radiant Energy System (CERES) sensors mounted on board the same platforms (Terra and Aqua)."

*Referee comment:*
p2. l33. "limitations". Limitations to do what?
*Author's response:*
We mean the limitation to resolve cloud and their radiative effect at smaller scales that the CERES footprint size. We will rephrase that sentence.
*Author's changes to the manuscript:*
The rephrased sentence will read "Limitations to resolve small scale clouds and their radiative effect might arise from the coarse spatial resolution of CERES (footprint size of approximately 30 km) and from the fact that the clear-sky fluxes are exclusively based on clear-sky pixels (and interpolation of clear-sky fluxes for gap filling on monthly scales), for which the spatio-temporal sampling is reduced and the meteorological conditions are likely to be biased."

*Referee comment:*
p3. l3-7. Please provide references for the WCRP GEWEX data and the ISCCP DX data.
*Author's response:*
Thank you. We will include the reference Stackhouse (2011) and Rossow and Schiffer (1990).
*Author's changes to the manuscript:*
Included reference Stackhouse (2011) and Rossow and Schiffer (1990).

*Referee comment:*
p3. l10-14. "based on the rationale above". It is clear why these data are required, compared to the MODIS/CERES and GEWEX data sets. However, what are the other data sets based on AVHRR lacking (PATMOS-x, CLARA-A2, Cloud-cci) that this paper will address with Cloud cci 3? A sentence on p2, line 19-21 would help clarify the shortcomings of those existing data sets.
*Author's response:*
Thanks for this suggestion. We will add the suggested sentence; however we find that sentence better placed on p3 l 13, as we need the provision of a full suite of cloud and radiation properties to make the point.
*Author's changes to the manuscript:*
We will add the following sentence near line 13 on page 3: "The availability of the full suites of cloud and radiative flux properties will also make these data superior to the already existing AVHRR-based datasets mentioned above."

*Referee comment:*
p3. l20-31. This paragraph pre-empts the findings ("superior") and methodology. The relevant information is better placed in section 2.1.
*Author's response:*
Thank you for this suggestion. We kind of see the point. However, algorithm developments (content of section 2.1) are only one reason for this finding. The longer time period covered and in particular the inclusion of the radiative broadband fluxes are strong contributors to this finding/conclusion as well. As these characteristics are motivated in the introduction we would like to leave that paragraph where it is right now.
*Author's changes to the manuscript:*
None.

*Referee comment:*
p4. l2. Please add that table 1 contains all abbreviations used throughout the text. CER had not been introduced in the main text prior to p6 and it took a moment to figure out its meaning.
*Author's response:*
Thank you. We'll do that.
*Author's changes to the manuscript:*
We will modify that sentence: "The set of cloud properties included in v3 is identical to v2 and is outlined in the upper part of Table 1, which also gives all cloud property abbreviations used throughout the paper."

*Referee comment:*
p4. l16. "much larger set". How do the two sets of training data compare? Did both v2 and v3 use CALIOP, but v2 just used fewer overpasses?
*Author's response:*
Yes. Both used CALIOP and for v2 we used much fewer overpasses compared to v3 (about a factor of 10). We will clarify in the text.
*Author's changes to the manuscript:*
The corresponding sentence will be changed to "The ANN for cloud detection (ANNmask) has been retrained using a much larger set of training data (approx. 10 times more collocation data used for v3 than for v2), which is composed of…."

*Referee comment:*
p6. l16. A "lower" CTP mean is not explained by more very low-level clouds, which have higher CTP. It appears that over the West Pacific and Maritime Continent, mean CTP has generally increased, which could be due to detection of more low-level clouds. Please re-consider this statement.
*Author's response:*
Thank you. This is indeed a type. A "higher" CTP it is. We will revise that sentence.
*Author's changes to the manuscript:*
That sentence will be changed to "Mean CTP is higher in v3 than in v2 in the Tropics…."

*Referee comment:*
p7. l1. Regarding the validation, did the authors consider performing the validation separately for daytime and nighttime observations? The algorithms use different channels and the authors consider nighttime COT and CER "experimental". It would be useful to understand the algorithm performance for different times of the day.
*Author's response:*
It is indeed useful. For this manuscript however we would prefer presenting just the general figures as the discussion would get to extended otherwise. Important to consider in this respect that there is a Project Validation Report soon being issued in which we plan to stratify the validation results wrt. illumination conditions. We will add this link to the text.
*Author's changes to the manuscript:*
At the end of Section 2.3 we will add the following sentence "An even broader assessment of the quality of the presented dataset can be found in PVIR (2019), in which the results are also stratified by illumination conditions among others."
Along with adding the reference:
PVIR – Product Validation and Intercomparison Report (PVIR) ESA Cloud_cci, 2019, Issue 6, Rev. 0; DD/MM/2019, available from http://www.esa-cloud-cci.org/?q=documentation_v3, 2019.

*Referee comment:*
p7. l3. Please, briefly explain how the collocation is carried out. In particular, what is the impact of the temporal mismatch between CALIOP and AVHRR? And what is the impact of the mismatch in footprint?

*Author's response:*
The collocations are done identically to Stengel et al. (2017). Most important facts are that all collocations are based on searching for the nearest neighbour in the Cloud_cci Level-3U data to each CALIOP observation. Given that a L3U grid box is usually of smaller size than 5 km, 5 km is the maximal allowed spatial mismatch. Most collocations have much lower spatial mismatches. Due to the similar orbital characteristics of NOAA-18 and NOAA-19 compared with CALIPSO, a very large set of collocations can be retrieved, which in turn also allows for very strict criteria and still infer a sound collocation database. In this context, the temporal mismatch criteria was set to a time window of±3 min. While the systematic deviations to CALIOP do not significantly depend on the match-up criteria, the random deviations do. We will summarize all this information in the text.

*Author's changes to the manuscript:*
We will add the following sentences: The collocations between CALIOP and the AVHRR-PM data were done as reported in Stengel et al. (2017) with the most important fact being that only those collocations were included for which the spatial and temporal mismatch was below 5 km and 3 minutes, respectively. These criteria were chosen as compromise between using best spatial and temporal matches, and allowing for a compositions of a sound data basis to be used in the validation. Important to note that the random deviations of AVHRR-PM to CALIOP depend on the defined criteria, while the systematic do most likely not.

*Referee comment:*
p7. l22-23. Why would improved identification of liquid clouds lead to reduced POD for ice clouds? This suggests that some ice clouds are now erroneously identified as liquid. Does that mean there are more "false alarms" in terms of liquid cloud detection?

*Author's response:*
That sentence is indeed misleading. For the v3 algorithm the probability of detection (POD) of liquid clouds is improved while the POD for ice clouds is slightly reduced, still leading to an overall increased (improved) HSS values.

*Author's changes to the manuscript:*
We will modify that sentence to: "Comparing the HSS score as overall measure for the correct cloud phase detection, v3 performs better than v2. The POD of liquid clouds is significantly improved in v3, while a small degradation in POD of ice clouds is found in v3 compared to v2."

*Referee comment:*
p8. l5-15. It would be helpful to consider the results from Tables 4, 5, and 6 through a visual comparison, as done in Figure 4. A scatter plot (or 2D histogram) of CTH, LWP, and IWP comparing the data set with the "truth" could help identify where biases are most likely to occur. For instance, the CTH bias of ice cloud could be mostly due to the highest clouds, even at high COT, as these might have a region of low extinction coefficient near cloud top, that would lead to higher CTH in CALIOP. A scatter plot could show this clearly. Similarly, LWP and IWP are highly skewed variables and the metrics presented could be affected by a few outliers. A scatter plot or 2D histogram (perhaps shown on a logarithmic scale) could indicate whether LWP and IWP estimates are typically good, or whether there is a consistent bias across cloud types of all LWP and IWP values.

*Author's response:*
Thank you for this suggestion. We have included 2d histograms for CTH, LWP and IWP.

*Author's changes to the manuscript:*
We have included 2d histograms for CTH, LWP and IWP.

*Referee comment:*
p9. p10. p11. Please rename standard deviation to "root mean squared error", which is presumably what is reported.

*Author's response:*
We actually mean the standard deviation of the error (which is basically identical to the bias-corrected root mean squared error). We will add to the captions of tables 4,5 and 6 that we mean the standard deviation of the error and the mean error.

*Author's changes to the manuscript:*
We will modify the captions of tables 4,5 and 6 to "….standard deviation of the error (Std), the mean error (bias) …."